# The demand for online grocery shopping: COVID-induced changes in grocery shopping behavior of Canadian consumers

**Viktoriya Galushko**[1]*, **Alla Riabchyk**[2]

**1** Economics Department, University of Regina, Regina, Saskatchewan, Canada, **2** Department of Marketing and International Trade, National University of Life and Environmental Sciences of Ukraine, Kyiv, Ukraine

* Viktoriya.galushko@uregina.ca

## Abstract

The COVID-19 pandemic has had a lasting impact on many economies around the globe. One area where significant changes have been documented is consumer behavior. A questionnaire survey was carried out to understand the impact of COVID-19 on grocery purchase behavior of Canadian consumers and evaluate the permanence of these effects. With a focus on online grocery shopping, this work integrates multiple existing theories of consumer behavior to explore the influence of different factors on consumers' adoption of online mode of grocery shopping during the pandemic and their intentions to continue the use of this mode in the post-pandemic world. A total of more than 600 usable survey responses were analyzed using statistical analysis and a Logit econometrics technique. The results reveal that 72% of the survey participants had to alter their grocery shopping habits as a result of the COVID-19 pandemic; 63% of these consumers claim that the changes that occurred would prevail in the future, with no return to the "pre-COVID normal". The results also show that the pandemic resulted in significant proliferation of online grocery shopping among Canadian consumers. Further, the findings show that the important factors that explain adoption of online grocery shopping and the shift towards higher reliance on online grocery purchases in the future include the perceived threat of COVID, pre-COVID shopping habits, socio-demographic characteristics, and the variables that capture technological opportunities and abilities.

## 1. Introduction

The impact of the pandemic on the economic, social, and psychological aspects of people's lives and the widespread adaptation of consumers to various constraints during the pandemic has spurred considerable interest among academic and market researchers. In the past two years, a body of literature emerged exploring pandemic induced changes in consumer behavior including how and where money should be spent [1]; stockpiling, impulsive and compulsive buying behaviors [2–4]; new demand for wellness products [5]; and proliferation of e-commerce [6]. However, the impact of the pandemic on grocery purchase behavior seems

**Data Availability Statement:** An Excel file called "S1 File Data" that accompanies this resubmission contains the dataset used for the analysis. S1 File Appendix provides a detailed variable description,

using the same variable names as in the data file so that the readers can easily replicate the results.

**Funding:** This research was supported by the Deans Research Award, provided by the Faculty of Arts at the University of Regina to Viktoriya Galushko. The funders had no role in study design, data collection and analysis, decision to publish, or preparation of the manuscript.

**Competing interests:** The authors have declared that no competing interests exist.

under-researched especially in relation to lasting changes in consumer's grocery purchase behavior. This study investigates the changes in grocery buying behavior of Canadian consumers during the pandemic as well as the potential longevity of these changes.

Food purchasing behavior underwent significant changes during the pandemic. While online purchases of non-food items were relatively common before COVID, food purchases were for the most part limited to in-store purchases with only a small proportion of grocery shopping taking place on-line. However, the coronavirus pandemic altered consumer preferences towards online grocery shopping–a promising but formerly niche industry. For example, one Canadian survey conducted in March 2020 at the onset of the pandemic indicated that 9% of Canadian consumers shopped for groceries on-line for the first time, which is a 6-fold increase from pre-pandemic levels (1.5%) [7]. Another survey administered by the Solutions Research Group indicated that the share of consumers in Canada who shopped online for groceries dramatically increased in 2020, with 17% of the sample respondents indicating on-line purchase of groceries in the week preceding the survey compared to just 5% reported in 2016 [8].

In many jurisdictions, including Canada, pandemic restrictions stayed in place for a relatively long period of time. Some psychologists claim that it only takes about 21 days for people to develop a new habit. [9] found that the average time it takes for a new behavior to become a habit is around 66 days, with individual times varying from 18 to 254 days. In their study they determined that it is the first days that are especially important in setting a foundation for success. Supermarkets responded rapidly to the COVID-19 lockdowns and restrictions by increasing their on-line order capacity and expanding order pickup options. If consumers who switched to on-line purchases had rather positive experiences, they could have discovered an alternative that is more convenient, affordable, and accessible. This increases the likelihood that on-line purchases would be utilized on a regular basis in the future, thus becoming a habit. It should also be mentioned that in some instances the stores were not ready for a surge in online grocery shopping. To meet the soaring demand, some supermarkets tried automating the picking of items for consumer orders but given the complexity of it reverted to manually picking customers' orders in stores. This impacted the capacity of supermarkets to fill orders in a timely manner [10].

The pandemic-induced changes in grocery shopping behavior, including adoption of online grocery shopping, highlights a number of important considerations that this study seeks to address:

- What is the nature of the changes in grocery shopping behavior of Canadian consumers that emerged during the COVID-19 pandemic?

- Has the pandemic induced a shift towards online grocery shopping that was a niche market before the crisis? What are the drivers of online grocery purchasing behavior?

- Will the observed changes in grocery shopping behavior, more specifically a trend towards increased grocery purchases through online platforms, persist into the future?

A careful look into the changes in grocery shopping behavior and the nature of the shift towards on-line grocery shopping is important to assess and possibly re-define the role of smaller retailers and family farms that may lack adequate resources to compete with large supermarkets in the on-line space. Documenting changes in grocery shopping behavior, understanding the factors that contributed to adoption of online grocery shopping platforms, and gaining knowledge of the changes brought about by the pandemic will provide important information for the retail food industry. More specifically, knowledge of whether the trend to buy groceries on-line will continue into the future should help supermarkets and grocery stores more effectively allocate their resources for marketing strategies. Better understanding

of the "new" consumer should help the retail industry make better informed investment decisions such as investments in infrastructure for fulfilment of on-line orders including in-store automation and innovation in marketing and e-commerce.

This paper integrates the findings from a number of theories of consumer behavior including the theory of planned behavior, technology acceptance model, technology continuance theory, health belief model, and motivation-opportunity-ability model to build an econometric model that explores the drivers behind adoption of online grocery shopping during the pandemic and consumers' intentions to continue to use this mode of shopping in the future. This work contributes to the existing scholarship in three ways. First, we provide an extensive descriptive analysis to document changes in grocery buying habits of Canadian consumers during the pandemic and expected future trends. Second, despite a global increase in online purchases since the start of the pandemic, there is still a lot of uncertainty in the existing literature around the drivers of online purchasing behavior, particularly online purchases of groceries [11]. Most of the existing studies that explore adoption of online grocery shopping focus on consumers' perceptions of the advantages and disadvantages [12, 13] or attempt to identify consumer traits of internet shoppers [14, 15]. As such, the existing literature mostly explores the importance of adoption constructs, where the use of online grocery shopping is the result of a conscious decision following the analysis of advantages/disadvantages. There is very limited literature on the role of unexpected situational factors—specific events in the lives of consumers that trigger a change—in adoption of online grocery shopping. The COVID-19 pandemic was a situational factor and this paper is an attempt to bridge the gap in literature and gain an enhanced understanding of consumers' response to the unanticipated shock caused by the pandemic.

Third, by analyzing decisions of Canadian consumers to adopt online grocery shopping in Canada during the pandemic and continue with online grocery purchases post-pandemic, this work creates knowledge about the "new" consumer that emerged out of the COVID-19 pandemic. This knowledge will benefit retailers, producers, and agri-food sector stakeholders, and allow them to adjust their operations and sales strategies to reflect the new reality of the post-pandemic marketplace.

The remainder of the paper is structured as follows. Section 2 provides a brief literature review, highlighting changes in consumer behaviors during pandemics with special consideration of the COVID-19 pandemic. Section 3 builds the theoretical model and develops propositions. The research methodology is described in Section 4. The results and discussion of this study's findings are presented in Section 5. Section 6 concludes the paper with policy implications, limitations of the study, and suggestions for future research.

## 2. Background literature

The existing literature identifies a number of factors that motivate consumers to adjust their buying behavior. These factors include economic recessions or slowdowns, social life changes, technological breakthroughs, changes in rules and regulations, natural disasters, and pandemics [16]. Since the emergence of COVID-19 in 2020, many studies set goals to investigate the impact of the pandemic on consumer behavior [17]. The existing literature has applied various behavioral theories to explore the observed adaptations in people's buying habits including panic buying, impulsive buying, digitalization of shopping, and shifts in consumer preferences.

During times of unexpected crises, people's ability to make rational decisions and judgements is significantly impaired, and buying patterns change as people anticipate shortages of essential goods and are faced with uncertainty of the duration of the crises [18]. Such irrational

consumer behaviors as panic buying and hoarding of necessities have been observed as one of the first changes in consumer behavior in response to many crises, including pandemics and natural disasters [19, 20]. For example, during the Ebola outbreak in the African countries in 2012, restrictions on people's movement and the establishment of quarantine zones led to panic buying and, as a result, food shortages [21, 22]. In 2003, during the severe acute respiratory syndrome (SARS) outbreak in China, social anxiety and fear led to panic buying of drugs, masks, and disinfectants as well as basic food items such as instant noodles and biscuits [23, 24]. Yet, during earlier disease outbreaks, panic buying behavior of consumers was not as wide-spread and pronounced as during the most recent COVID-19 pandemic, with consumers worldwide flocking to local stores emptying the shelves for long shelf-life food, hand sanitizers, masks, and toilet paper.

[25] highlight that it was the uncertain circumstances surrounding the COVID-19 pandemic that increased fears among consumers and subsequently left them susceptible to herd mentality behaviors where people tend to follow the crowd rather than consider a more logical, individualized approach. From a psychological response point of view, panic buying could have been triggered by an underlying conflict between desire to maintain routines and the uncertainty of duration of the pandemic, by an inner "self-preservation" mechanism to cope with a stressful unmet situation that was aggravated by constant news of rising numbers of infected individuals and deaths, or by consumers' feelings of loss of control over their situation [26]. Drawing on the theoretical contributions of the health belief model, perceived scarcity, and anticipated regret theories, [27] noted that when faced with the threat of a disease outbreak, consumers engage in protection motivation behavior. In the context of the COVID-19 pandemic, such behavior resulted in panic buying as the first response to the disease outbreak to prevent a shortage of essential products. Buying products in large quantities also served as a method of protecting consumers from contracting COVID-19 by reducing the frequency of shopping trips and exposure to external environments, thus reducing their susceptibility to contracting COVID-19. Similar to previous pandemic events, panic buying during the COVID-19 pandemic had a temporary nature and purchases stabilized within a couple of months after the announcement of COVID-19 a global pandemic [25].

The literature on consumption behavior during crises also reveals that individuals generally suffer negative emotions and use consumption as a strategy to cope with these emotions [28–30]. However, economic recessions and slowdowns accompanied by a loss of purchasing power, force consumers to seek stability by reducing consumption levels, buying cheaper goods, seeking bargains, and prioritizing purchases of essential products while forgoing non-necessities and not immediately required expenses [31–35]. In other types of crises such as natural disasters and pandemics that are associated with an increased level of stress and emotional imbalance rather than changes in affordability, consumers can respond differently and actually increase their consumption levels and adopt abnormal buying behaviors such as impulsive and compulsive purchasing [36, 37].

The COVID-19 pandemic caused an unprecedented disruption to peoples' lives and economies worldwide. In the early months of the pandemic, millions of people around the world lost their jobs; in Canada, as lock-down measures and stay-at-home orders were implemented throughout the country, nearly 2 million jobs were lost in April 2020 and the unemployment rate soared to 13% compared to the pre-pandemic unemployment rate of 5.6% in February 2020 [38]. In addition to economic hardships that arouse at the onset of the pandemic, people's lives were exposed to fears for their own health and wellbeing and that of their family members. The pandemic-induced disruptions to normal ways of living also resulted in major lifestyle changes and significantly increased levels of stress, anxiety, depression, and psychological distress [39, 40]. As consumers were faced with multiple waves of COVID-19, the first stage

response that manifested itself in hoarding behavior led to a second stage response of coping [11].

After adjusting their thinking and decision making to attenuate the pandemic-induced constraints, consumers significantly changed their shopping behavior in a variety of ways. These changes were not limited to consumption levels, but also included changes in the way consumers shop. Recent studies have documented increased tendency of consumers to engage in impulsive buying behavior [41–43] and shift demand for certain products due to changes in preferences including hygiene products or healthy foods [5, 44–49]. To adjust to a "new normal" and sustainable way of life during the pandemic, consumers responded by changing the ways they shop including who shops, how they shop (in store or on-line), and where they shop (busier large stores or less busy smaller stores). Drawing upon protection motivation theory (PMT), temporal construal theory (TCT), and self-determination theory (SDT), [50] presents the narratives of UK consumers highlighting how self-control with respect to consumer thoughts, emotions, and behaviors encouraged some consumers to change from shopping at larger supermarkets to smaller, local convenience stores or shift to shopping online where, in the safety of their own home, they felt greater control over "a safer shopping environment."

It is widely accepted that the online sector is crucial in providing vital access for customers to essential products and the expansion of online shopping that occurred during the recent COVID pandemic was a notable change [11, 51–54]. The literature demonstrates that expansion of e-commerce was also observed during previous disease outbreaks [55, 56]; for example, the SARS outbreak in China is believed to have been a key driver behind China's subsequent emergence as the world's largest e-commerce market [57]. During the COVID-19 pandemic there was an unprecedented surge in online retail demand, with some sources suggesting that in 2020 the share of e-commerce in retail sales grew at two to five times the rate before COVID-19 [58]. While on-line purchases of non-food items became common in many countries and jurisdictions even prior to COVID-19, grocery shopping for the most part remained limited primarily to in-store visits. In many countries consumers shifted their grocery purchases online despite the fact that buying food is associated with sensory attributes. [51] explored the change in demand for on-line food shopping in Taiwan using the data from Ubox and found that the variety of food products purchased on Ubox significantly increased due to COVID-19. In the UK, while it took two decades for online grocery sales to increase from 0 to around 7% of total grocery sales, the country experienced an increase from 7% to 13% in just about eight weeks following the onset of the pandemic [10]. In Germany, analyzing an extensive panel dataset of 17,766 households, [59] found that volume-based share of online grocery purchases increased from around 0.6% to almost 1.2% at the onset of the pandemic; [60] reported that, at the time of their survey in October/November 2020, 72% of the survey participants had experience with online grocery shopping and more than half of them used the service for the first time during the pandemic. To the best of our knowledge, there is no research study that explores how grocery purchasing behaviors evolved during the pandemic in Canada. As such, this study is an initial effort towards bridging this gap in literature by investigating the shifts to online grocery shopping in Canada during the pandemic and whether this represents a permanent shift toward online grocery shopping for Canadian consumers.

## 3. Consumer adoption of online grocery shopping: theoretical model and development of hypotheses

Past literature has investigated consumers' grocery purchase behavior to identify the factors that predict consumers' adoption of online grocery shopping [61–64]. Several research

perspectives have been suggested, including the theory of reasoned action and the theory of planned behavior [65], the technology acceptance model (TAM) [66, 67], the theory of adoption of innovations [68, 69], the perceived risk theory [70], and others. In our analysis below, the findings from these various theories are used to build a theoretical model to facilitate the econometric analysis and provide a better understanding of the factors influencing consumers' decisions to alter their grocery shopping behavior during the COVID-19 pandemic.

In the current context, grocery shopping/buying behavior is a broad concept that would include but not be limited to such observed variables as frequency of in-store visits, the preferred mode of shopping (e.g. brick-and-mortar or online), the preferred sources of meals (e.g. restaurant ready-to-eat items that do not require grocery-store purchases vs home-made meals from items purchased at grocery stores), and the preferred shopping venue (e.g. large supermarkets vs small convenience stores). Our focus in this study is on the adoption or expansion of online grocery shopping, as one aspect of adaptations in consumers' grocery purchase behavior during the pandemic.

### 3.1. Perception of risks and demand for on-line grocery purchases during the pandemic

Consumption behavior during pandemics or periods of crisis is determined by consumers' perception of risks including social, physical, financial, and psychological and constraints imposed by governments [46]. During pandemics, staying healthy often becomes one of the most important basic and psychological needs of consumers. The protection motivation theory (PMT) [71] and health belief model (HBM) [72] have been commonly applied in the literature to understand how consumers alter their behavior in response to external factors that are believed to pose risk to their health [73, 74]. Since the COVID pandemic began, these theories have been extensively used to study abnormal buying behavior of consumers, more specifically panic buying [27, 75]. Outside of panic buying, which is considered to be irrational consumer behavior, these theories can be useful in exploring adaptations in buying behaviors in general. According to these theories consumers would change their behavior to protect themselves from a threatening event based on their perception of four considerations: severity of the dangerous situation, likelihood of the incidence of danger, benefits of the suggested preventive behavior, and personal ability to adopt the behavior [27]. Perceived susceptibility and severity (i.e., perceived vulnerability to and risk of contracting COVID-19) are directly related to consumers' level of worry and task, and response orientation [27].

We can, therefore, formulate the following proposition:

**Proposition 1:** *Consumers are motivated to change their buying behavior, more specifically adopt online mode of shopping, to reduce the possibility of exposure to COVID-19 when they perceive the susceptibility to and severity of the disease to be high.*

According to HBM, perceived susceptibility is defined as an individual's belief about their chances of getting a certain condition, which is coming in contact with the coronavirus in the context of the COVID pandemic. Perceived severity refers to a consumer's belief about the seriousness of the disease if they get ill. For a consumer to change their grocery buying habits, they should believe they are at risk for illness and/or negative health outcomes if they continue their current behavior. The opposite is also true: when people believe there are no threats to their own or their family members' health, they tend not to take protective measures and no adjustments to their shopping behaviors are made.

Perceived COVID risks can be influenced by a number of factors. One of the important factors in framing consumers' beliefs about the situation is knowledge. Knowledge is

accumulation of facts gained from various sources including internet, media, and an individual's experiences. The narratives from the media and information about the number of COVID cases and deaths in the community where the individual resides can ease or aggravate COVID-related fears. While the federal government in Canada had full control of the COVID-related measures with regards to international travels, it was within the jurisdiction of the provincial governments to collect and report the COVID-related statistics and implement the within-province measures to curtail the spread of the virus. In the local news reports, each province focused on the epidemiologic situation in that particular province and we stipulate that it is the "local" COVID summary that many consumers were putting more weight on in deciding if any adaptations in their behavior were warranted. Therefore, the province of residence could serve as a proxy for consumer information about the COVID situation and can to some extent capture differences in perceived COVID threat and as a result play a role in influencing grocery purchase behavior. A few studies have shown that people in the neighborhoods with higher number of positive COVID-19 cases were more likely to replace some of their in-store purchases with online ones [76, 77]. This result from the existing literature leads us to the following proposition:

**Proposition 1a:** *Consumers residing in provinces with a higher number of deaths from COVID-19 were more likely to experience bigger fears and, therefore, more likely to shift part or all of their grocery purchases online.*

While the flow of information is essential to gain knowledge, the ability to understand the issue from many different perspectives is also important in the formation of beliefs about susceptibility and severity. We hypothesize that educational attainment plays a role in how much information is received and how this information is processed and transformed into "knowledge".

Perceived susceptibility and severity reflect subjective risk. Health psychologists have found that people often have a tendency to believe that negative events, such as becoming ill or finding oneself in a situation with a negative health outcome, are more likely to happen to other people than to themselves [78, 79]. While the media throughout the world portrayed COVID as a very high risk illness, reactions of the public varied significantly, from denial of the severity of COVID, to panic buying of toilet paper and altogether avoiding any contact with people outside the household [80, 81]. Personality traits can play a role in shaping a person's perceived susceptibility and severity [82]. We postulate that a person who is a "worrier" by nature–someone who has a more negative thinking style and higher anxiety levels in risky situations, would consider COVID-19 a bigger threat and would be more motivated to use the online mode of grocery shopping as a safety and preventative measure.

**Proposition 1b:** *"Worriers" by nature are more likely to consider COVID-19 a bigger threat and, as result, more likely to adopt/increase online grocery shopping to avoid exposure to the virus.*

Fears of catching the virus and concerns about an individual's health if the individual contracts the virus are also expected to vary across individuals depending on personal experiences with the virus (for example, a severe COVID case or COVID-related death among close contacts or family members) or the individual's personal risk factors such as gender, age, and/or the presence of comorbidities. Some studies have shown that the average fear of COVID-19 was higher in women than in men [83, 84]. Research also showed that the risks of developing health complications from COVID increased with age; in the U.S., about 81% of deaths from the disease were in people age 65 and older. This information about age differences in patients with COVID-19 was presented in the media and could therefore be used by consumers in shaping their COVID risk perceptions.

Household composition, more specifically the presence of children in the household, can also affect perception of risk during the pandemic. The existing literature has found that people living in families with children had higher risk perception as they were concerned about their children getting infected with COVID-19; as a result, they were willing to put in more effort to reduce their exposure to the virus [85].

## 3.2 Consumer's socio-economic background and demand for online grocery shopping during the pandemic

The Motivation-Opportunity-Ability (MOA) model, originally proposed by [86] and later applied to online grocery shopping behavior by [62] links the advantages (e.g. independence of the store opening hours) and disadvantages (e.g. the inability to ascertain the quality or return the product) of online grocery shopping to different online shopping opportunities and as a result highlights the motivations of consumers to adopt it. [62] notes that MOA can be affected by socio-demographic factors. For example, household composition is likely to affect the opportunity cost of time. In households with all adults having full-time jobs, designing time-related shopping strategies that fit into the schedule of all working adults may be a challenge, in which case, the flexibility of online shopping in terms of time and space may offer additional motivation to utilize this mode. In the same vein, if a household has small children, the time may be more scarce as parents usually have to allocate a fair amount of time to children related activities; as a result, the opportunity cost of time will be higher and the motivation to save time through online grocery shopping may be much stronger than for a household with no children or all grownup children. [87] found that the group of shoppers that was especially positive about and appreciative of opportunities presented by online grocery shopping was mothers with young children.

COVID-19 brought a dramatic change in the lifestyle of people [5], and it is very likely that people belonging to different socio-economic backgrounds experienced different effects and adjusted their grocery buying behavior in different ways. Changes in income influence affordability [5], which in turn can motivate consumers to alter their shopping behavior. Furthermore, online grocery shopping is often associated with additional delivery charges, thus creating perceptions for lower income groups that online grocery shopping is costlier and therefore less useful. It has been reported that higher income households do almost three times as much online shopping as their lower-income counterparts [88].

The pandemic uniquely affected families with minor children by significantly disrupting their routines, changing relationships and roles, and altering usual child care, school, and extra curriculum activities [89]. As many parents tried to handle work and child care responsibilities during the pandemic and lived a life of isolation with no playdates or children activities [90], it may be the case that the pre-pandemic shopping habits couldn't be easily sustained and alternatives to in-store grocery shopping would have been sought. Also, in Canada, government officials and store management urged and in some cases made it the policy not to bring children to stores, including grocery stores. The above factors may have played a role in triggering changes in grocery shopping format.

Based on this, we posit the following propositions:

**Proposition 2**: *Change in grocery shopping behavior, and, in particular, new or increased demand for online grocery purchases, is significantly associated with socio-economic background.*

**Proposition 2a:** *Due to additional delivery charges associated with online orders, perceived cost of online grocery purchases is higher for individuals who have lower income, thus making online grocery purchases less attractive.*

**Proposition 2b:** *Changes in income due to the pandemic significantly influence shopping behavior, including new or increased demand for online grocery purchases.*

**Proposition 2c:** *The presence of small children in a household is significantly associated with life-style changes of consumers during the pandemic and is therefore an important determinant of changes in grocery shopping behavior, including new or increased demand for online grocery purchases.*

## 3.3 Personal abilities to alter grocery shopping behavior and demand for online grocery purchases during the pandemic

Based on the theory of planned behavior [91], consumer behaviors are influenced by behavioral intentions and one of the determinants of the latter is perceived behavioral control. Perceived behavioral control represents an individual's perception of how easy or difficult it will be to perform the behavior of interest and is closely related to self-efficacy–one's belief about their own ability to perform a certain behavior by employing their own skills [92]. Self-efficacy, in turn, will influence consumer's perceived ease of use–a central concept in TAM. In the context of online grocery shopping, we postulate that self-efficacy is affected by technological abilities and opportunities to engage in online shopping and grocery shopping habits prior to the pandemic.

**3.3.1 Technological opportunities/abilities and demand for online grocery shopping.** Since online shopping requires certain skills and resources (Internet, computer, cell phone), perceived ease or complexity of online grocery shopping can play a pivotal role in determining perceived behavioral control and self-efficacy. When adaptation in consumer behavior requires use of information technologies (internet, computers), TAM is the most widely used model to understand the factors driving consumers' choices. This model suggests that a person's behavioral intention to use a technology is determined by perceived usefulness and perceived ease-of-use. Ease-of-use–the extent to which a consumer believes that online grocery shopping is free of effort–will be determined by technological abilities and opportunities of a consumer.

Research suggests that behavioral intention, adoption, and acceptance of technology is moderated by demographic characteristics, including age, gender, income, and education [93–95]. It has been found that older adults have low computer and internet self-efficacy and often believe that they are too old to learn a new technology (new ways of shopping in the current context) [96]. The existing literature has also shown that younger adults generally have lower level of computer anxiety and are therefore more likely to engage in opportunities where information systems skills are required such as online shopping [97]. Some studies have provided support for the moderating role of gender in TAM context, however, the results in the literature are mixed [97]. The decision to switch grocery shopping online is likely to require some knowledge and research studies reveal that more educated individuals tend to have better abilities to grasp new information quickly than their less educated counterparts, and, as a result, there exists a positive association between education level and perceived ease of use of new technologies [94, 98]. People of lower socio-economic status (i.e. consumers with lower incomes) can perceive access to information technologies (computers, cell phones) and internet as more costly [94].

Therefore, we can formulate the following propositions:

**Proposition 3:** *Age is negatively associated with perceived ease of use of the Internet (technological ability) and, therefore, older adults are less likely to switch their online grocery purchases online.*

**Proposition 4:** *Perceived ease of use of computer technologies (i.e. technological ability) is higher for consumers who are more educated; therefore, individuals with higher educational attainment are more likely to switch their grocery purchases online.*

**Proposition 5:** *Technological ability to perform online grocery shopping is higher for consumers with higher incomes due to lower perceived cost of information technologies (access to computer, access to Internet); therefore, higher income consumers are more likely to switch online.*

**3.3.2 Pre-pandemic grocery shopping habits and ability to adopt new behavior.** Consumers generally know their own needs and respond to them by developing certain grocery shopping habits. The ease of adjusting to a particular situation in life for a consumer and the ability to adopt new buying behavior will be determined by the current behavior that is assumed to attend to the consumer needs: the bigger the gap between the current behavior and the new behavior, the less likely the consumer will choose to make the adjustment to adopt this new behavior.

Consumers perceive grocery shopping differently. While for some groups of consumers grocery shopping is just another weekly chore they would be happy to avoid, for other groups of consumers shopping for food is considered a fun activity enhancing social interaction and bringing the family together. When grocery stores are visited frequently as part of a fun leisure activity, then consumers can also find online grocery shopping as a fun activity and enjoy more shopping for groceries online during the pandemic. Some consumers place high value on freshness of fruits and vegetables and want them just the right ripeness. Such consumers are likely to be less inclined to switch their grocery purchases online and will instead prefer daily grocery store visits where they can inspect fresh produce and pick exactly what they want to eat [99]. Frequency of grocery store visits prior to the pandemic can serve as a proxy for an individual's utility derived from grocery shopping trips and/or value placed on freshness of produce.

The importance of specific grocery items in overall meals can also be an important feature of pre-COVID grocery shopping behavior. When a family meal plan involves buying a fair amount of fresh produce (fruits and vegetables) and fresh meats, consumers may be less likely to switch to online grocery shopping as this food category has been found to have the lowest overall satisfaction rate with consumers using online grocery shopping [100]. On the contrary, consumers who have a large portion of their meals originating from restaurants or fast food places may not require lots of fresh produce from grocery stores and as a result will be more satisfied with online purchases.

Consumer preferences for the type of grocery items they need, the quality, and selection of those items, are likely to be reflected in the chosen shopping venue. For example, large supermarkets offer a much bigger selection of items than local convenience stores. During the COVID pandemic, large supermarkets and smaller convenience stores responded differently to the restrictions imposed by the governments to curtail the spread of the virus; also, there was a substantial difference in online order fulfilment capacity between large supermarkets and smaller stores. Therefore, we hypothesize that the chosen shopping venue prior to the pandemic had an influence on the ability of consumers to adopt new grocery buying behavior, including shifting more of their purchases online.

Following the discussion above, we formulate the following propositions:

**Proposition 6:** *Consumer grocery shopping habits prior to the pandemic, including frequency of in-store grocery store visits, importance of groceries in family meals, the choice of shopping venue, and prior online grocery shopping experience, will significantly influence the ability to*

*make changes to the current behavior as a result of the pandemic and will therefore affect the likelihood of switching grocery purchases online.*

## 3.4 Psychological predisposition to adoption of e-commerce

The existing literature has shown that gender and age are two important descriptors of psychological pre-disposition to shop online. More specifically, prior research has shown that women shop more online than men, and younger people shop more online than older people [101, 102]. While the age and gender differences in online shopping can be explained from a viewpoint of TAM discussed above, some studies note that men and women are evolutionary predisposed to different shopping styles and it is these differences in male and female psychology that drive the observed differences in adoption of online shopping [103]. Therefore, we can postulate the following hypothesis:

**Proposition 7:** *Due to psychological differences, gender and age are significantly associated with new demand for online purchases.*

A summary of the above discussion on the determinants of adoption of online grocery shopping is presented in **Table 1**.

## 3.5 Intentions to continue online grocery purchases: Will the changes that emerged during the pandemic persist in the future?

Most of the existing studies on adoption of online grocery shopping focus on consumers' perceptions of advantages and disadvantages [12, 13] or attempt to identify consumer traits of internet shoppers [14, 15]. This literature assumes that the switch to online shopping occurs as a result of a rational thinking process where consumers carefully evaluate the benefits and

**Table 1. The key hypothesized determinants of "new" and increased existing demand for online grocery shopping.**

| Determinants | Expected impact |
| --- | --- |
| Personality type | "Worriers" by nature are more likely to adopt online grocery shopping. |
| Fears of contracting the virus | Higher fears are associated with higher likelihood of adopting online grocery shopping |
| Fears of developing complications from illness | Higher fears are associated with higher likelihood of adopting online grocery shopping |
| Age | Ambiguous. Older individuals are hypothesized to have higher perceived COVID risk; yet, older individuals are hypothesized to have lower technological abilities. |
| Presence of small children | Presence of small children is associated with higher likelihood of adoption of online grocery shopping both due to higher perceived COVID risks and due to more substantial life-style changes. |
| Frequency of in-store grocery store visits prior to COVID | Ambiguous; If higher frequency is an indicative of a high value being placed on freshness of products then it is less likely that in-store visits will be replaced by online purchases. However, if it is an indicative of "love" for grocery shopping, then online grocery purchases can increase. |
| Importance of specific groceries in meals | Higher importance is associated with lower likelihood of switching grocery purchases online. |
| Education | More educated individuals are hypothesized to have higher technological abilities and opportunities and are therefore more likely to adopt online grocery shopping. |
| Income | Higher income individuals are hypothesized to have higher technological abilities and the likelihood of adoption is higher |
| Sex | Ambiguous; the results from the literature are mixed. |

costs of adoption of online shopping methods and are therefore unlikely to revert back to brick-and-mortar. When the switch to online is caused by situational factors, such as the COVID-19 that unexpectedly disturbed people's lives, this switch may be just a temporary coping strategy rather than the result of a decision based on the advantages and disadvantages unrelated to the situational factors. For example, when [13] administered their survey before the pandemic, avoidance of stress and physical contact with others was mentioned by some respondents as a factor in doing grocery shopping online, albeit it didn't bear too much weight among the 51 identified benefits of e-shopping. The study by [13] revealed that cost savings associated with on-line shopping, both in terms of money and time, and greater availability of products online were mentioned most frequently as benefits of on-line grocery shopping. As long as there was a perception that online shopping saves time, favorable attitudes towards online shopping were formed. Functional motives such as saving money and time were found to be highly relevant in other studies as well [104–106]. For many consumers during the pandemic–a situational factor—maintaining social distancing was key. As such, one could hypothesize that a switch to online grocery shopping or increased tendency to buy online could have initially been triggered by the desire to limit contact with others, while cost and time savings could have come as a realized benefit later on for those who tried online grocery shopping for the first time during the pandemic. [64] find that situational factors such as health problems, a change in family circumstances, or promotional flyers usually act as triggers for the first use of e-grocery services. Over time, this initial exposure to on-line shopping seems to help consumers realize that they could actually benefit from using it, thus stimulating consumers to continue with e-grocery services. Some literature, however, notes that when the change is caused by situational factors, the adoption of online grocery shopping is an erratic process, driven by circumstances rather than by a cognitive elaboration and decision [61, 107]. Based on this literature, the adoption of online shopping is likely to be discontinued when the initiating circumstances change. [108] argue that consumers will be motivated to shop online only during the pandemic crisis and once the pandemic pressures subside online retailing will decline.

From a viewpoint of the theory of adoption of innovations, the diffusion process will be determined by the nature of innovation and, in a broad sense, innovations can be classified as continuous, dynamically continuous, and discontinuous [109]. Shopping for groceries online is arguably a discontinuous innovation [69, 107] as it requires consumers to significantly alter their behavior especially when one considers online purchase of items that are rich in sensory attributes such as fresh vegetables, meat, and fish. Therefore, one can expect this mode of grocery shopping to be discontinued beyond the pandemic.

Given that it is generally unclear whether consumers will perceive online-grocery shopping as continuous or discontinuous innovation and whether the realized benefits of online grocery shopping will outweigh the importance of sensory attributes of grocery items, it is worthwhile examining which factors influence consumers' intentions to continue increased online purchases of groceries.

A number of the existing theories of consumer behavior can be used to analyze intentions of consumers who increased or adopted online grocery shopping during the pandemic to carry on with this behavior into the future. The theory of reasoned action and planned behavior postulate that consumer behavior is determined by the consumer's behavioral intention, which, in turn, is a function of consumer's attitude, subjective norm, and perceived behavioral control [65, 110, 111]. In the context of online shopping where user acceptance of information technology (IT) is required for the action (actual online purchase), TAM was successfully applied in the literature. Stemming from the theory of reasoned action, TAM hypothesizes that consumers' attitudes are formed on the basis of perceived

usefulness and ease of use. Although TAM has been applied to examine continuance and post-adoption behavior [112, 113], the general feeling in the research community was that it lacked explanatory power and, as a result, two other models were developed to shift attention from initial acceptance to continued use—the Expectation Confirmation Model (ECM) and the Cognitive model (COG). In ECM, user satisfaction is the driving force behind individual's intention to continued use. COG states that continuous behavioral intention is determined both by satisfaction and attitudes and most studies have noted that the two concepts have to be considered as conceptually distinct with satisfaction being a transient and experience-specific affect and attitude being relatively more enduring transcending all prior experiences [114]. In the Technology Continuance Theory (TCT), developed by [114], the constructs from TAM, ECM, and COG have been integrated into one enhanced model of IT continuance. The two central constructs in TCT include satisfaction and attitude, and the first level antecedents include confirmation, perceived usefulness, and perceived ease of use. Adapting the TCT model to our survey data discussed in the next section, we can formulate the following propositions:

**Proposition 8**: *Technological abilities are significantly associated with future use of online grocery shopping. Technological abilities will be determined by age and income.*

**Proposition 9**: *Pre-COVID habits are significantly associated with the intention to continue "new" online grocery shopping behavior.*

**Proposition 10:** *Positive experience with online grocery shopping will positively affect satisfaction, and therefore, intention to continue "new" online grocery shopping behavior.*

**Proposition 11:** *Perceived benefits and perceived disadvantages of online grocery shopping will affect intention to continue using this mode in the future.*

**S1 File** presents a detailed description of the variables discussed above and used in this study.

## 4. Research methodology

### 4.1 Survey design

While we came across several studies that used questionnaires to estimate the importance of various factors in triggering adoption of online shopping platforms, we could not find any suitable questionnaire in the existing literature that would adequately capture the hypothesized theoretical model as discussed above. In developing the survey, we took cognizance of various items in the literature on the impact of COVID on consumers' buying and shopping behavior, which were found relevant for our study. In addition, we used various reports in electronic and social media that documented the changes in consumers' grocery shopping habits. First, we framed an open-ended questionnaire to gain a broad insight into the impact of the pandemic on grocery purchase behavior; this open-ended questionnaire was used to collect qualitative responses from our own networks of friends and colleagues. The received responses were used in the design of the close-ended questionnaire; the complete questionnaire can be obtained from the authors upon request.

The first section of the questionnaire gathered information on the socio-demographic and economic characteristics of the respondents. Part of this section was also intended to elicit information on perception of COVID-19 risks. The second section contained questions relating to the grocery shopping behavior prior to the start of the pandemic including frequency of in-store visits, preferences for grocery shopping venues, the role of online grocery purchases in total grocery purchases, and others; this section also gathered information on the changes in

grocery shopping behavior that occurred during the pandemic and the factors that triggered these changes. The third section of the questionnaire focused on online grocery shopping–the factors that played an important role in switching in-store purchases to online, benefits of and challenges associated with online grocery shopping, intentions to continue online grocery purchases, and other aspects of online grocery shopping. It is important to note, that although the survey was administered at one point in time (January 2022), the survey questions explicitly asked participants to describe their grocery shopping behavior prior to the pandemic, at the start of the pandemic, and at the time of the survey (i.e. almost 2 years into the pandemic). Therefore, the survey questions elicited changes in grocery buying behavior that were induced by the pandemic.

Prior to soft-launch of the questionnaire, a few colleagues with extensive consumer behavior research expertise were consulted to ensure adequate and appropriate coverage of the items as well as to provide their expert opinion for evaluating the ease of understanding the questions from the perspective of potential respondents.

## 4.2 Data collection and representativeness of the study sample

This study was approved on ethical grounds by the University of Regina's Ethics Board on October 22, 2021 and the online survey was administered by Qualtrics in January of 2022. To incentivize participation in this study, the Qualtrics survey team used their own reward strategy that varied across the individuals, with participants receiving various amount of award in the form of award points. Prior to participants' being allowed to take the survey, a preamble to the survey was provided outlining the purpose of this research, the survey procedure, and the participants' rights to withdraw at any time by just closing the browser, with the incomplete responses to be removed from the analysis. The preamble also emphasized the fact that the responses will be kept confidential, in compliance with the requirements of the Ethics Board. Moreover, since the survey was run by a third party, Qualtrics, the authors had no access to the information that could be used to identify individual participants during or after data collection. To obtain the participants' consent to use their responses in our study, the preamble concluded with the following statement: "*By completing the survey, you are giving your informed consent to participate in this study*".

Responses that were found to be incomplete and internally inconsistent were removed from the analysis. We also eliminated those respondents who took less than 5 minutes to complete the survey as this was sufficient evidence to suggest that they were very likely speeding through the survey, meaning the questions were not read carefully and answered thoughtfully. Straightlining–when a respondent chooses the same answer choice, e,g, the first answer option, over and over again,and inconsistent responses in surveys with survey time of less than 5 minutes were clear evidence that the respondents were either dishonest or careless. After cleaning the survey data, this study produced 651 usable survey responses.

Participants for this study were selected based on a number of characteristics. Eligible respondents had to be over the age of 18 and classified as the household's primary shopper responsible for at least 50% of the total household's grocery purchases. To ensure the representation from all Canadian provinces, a disproportional quota sampling technique was used. In addition, proportionate quota sampling was applied based on the income variable to ensure that the study sample closely resembles the target population, with income distribution mimicking the information on total income distribution in Canada in 2019, provided by Statistics Canada in Table 11-10-0238-01 (formerly CANSIM 206–0051) (**Figs 1** and **2**).

58% of the participants in our sample are females and 42% are males, which is very similar to population proportions. **Fig 3** shows that our sample is characterized by slightly higher

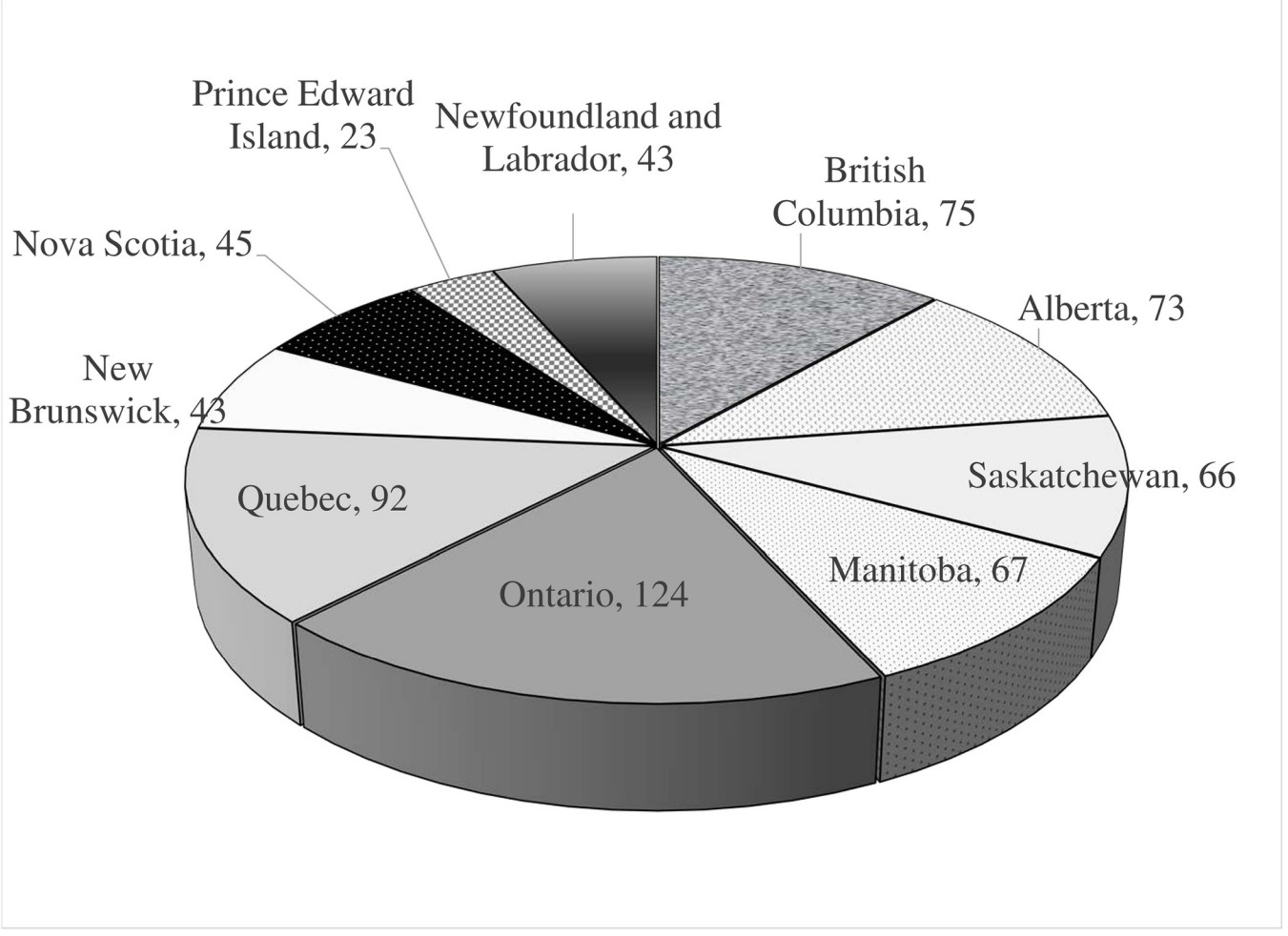

**Fig 1. Number of respondents in the study sample, by province.**

educational attainment than the target population and as **Fig 4** illustrates, the study sample closely resembles the population with regards to age distribution.

### 4.3 Survey questions and the study variables

This study attempts to understand the factors that explain the adoption of online grocery shopping during the pandemic (Model 1) as well as the factors explaining future use of online grocery shopping platforms (Model 2). The respondents were asked to classify themselves as (1) first time online grocery shoppers, (2) on-going online grocery shoppers, and (3) not online grocery shoppers. The first group includes those who started shopping for groceries online only as a result of COVID-19 and represents new demand for online grocery shopping; the second group (ongoing online grocery shoppers) includes those who shopped for groceries online prior to the pandemic as well as during the pandemic and represents the demand for online grocery shopping that existed before the pandemic; the third group includes those who, at the time of the survey, indicated they had never purchased groceries online The ongoing online grocery shoppers were also asked if the COVID-19 pandemic resulted in increased purchases of groceries online.

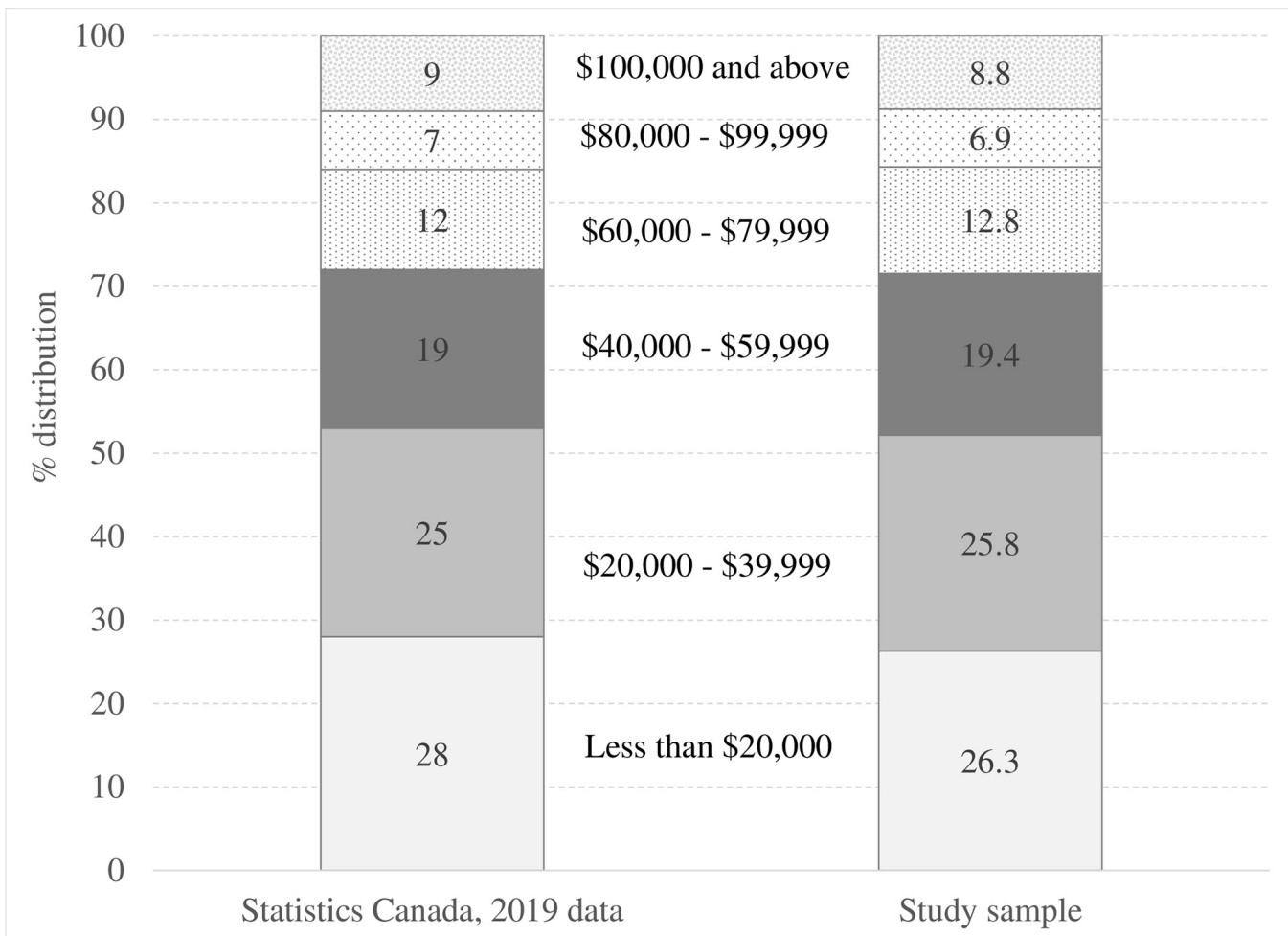

**Fig 2. Distribution of the respondents by income, Statistics Canada versus the study sample.**

In Model 1, the dependent variable captures adoption or increased use of online grocery shopping during the pandemic and is defined as:

$$y_1 = \begin{cases} 1 & \text{if first}-\text{time online grocery shopper} \\ 1 & \text{if an ongoing online shopper \& online purchases increased} \\ 0 & \text{if not an online grocery shopper} \\ 0 & \text{if an ongoing online shopper \& purchases didn't increase} \end{cases} \tag{1}$$

In Model 2, the dependent variable captures the intention of first time online grocery shoppers to continue using online grocery shopping platforms post-pandemic or an intention of on-going online shoppers to sustain increased levels of online grocery shopping in the future. To construct a variable that is relevant only for those consumers who reported purchasing groceries online at the time of the survey (i.e. first-time online shoppers and ongoing online shoppers) a number of questions from the survey were used. We inquired if the first-time online grocery shoppers thought they would continue buying groceries online in the post-pandemic times. The ongoing grocery shoppers who increased online purchases of groceries during the pandemic were asked if these increased purchases would continue into the future even when

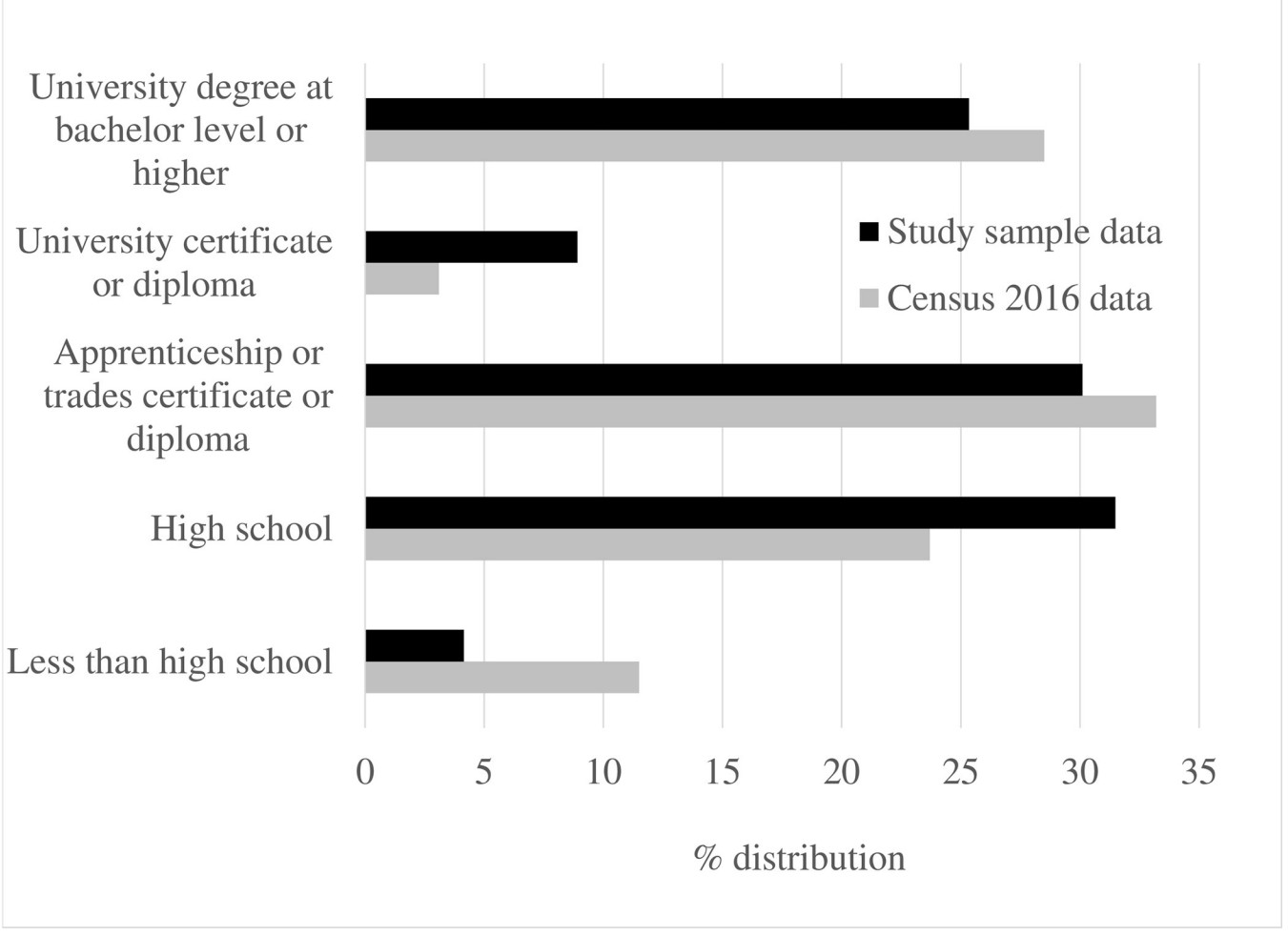

**Fig 3. Distribution by educational attainment, study sample versus census data.**

the pandemic is over. Therefore, for Model 2, the dependent variable is defined as:

$$y_2 = \begin{cases} 1 & \text{if first\(-\)time online grocery shopper\&online purchases will continue} \\ 1 & \text{if an ongoing online shopper\&increased online purchases will continue} \\ 0 & \text{Otherwise} \end{cases} \quad (2)$$

The validation of the conceptual models was performed using statistical testing and econometric analyses, with the results presented in the following sections. Given the dichotomous nature of the dependent variable, the choice for the econometrics estimation is the logit model with robust standard errors that was estimated by maximum likelihood estimation in Stata version 18. Logistic regression analysis is a standard econometrics estimation procedure that has widely been used in Economics to study adoption of e-commerce [115].

## 5. Results

### 5.1 Has the COVID-19 pandemic altered grocery shopping behavior?

Our results clearly show that the pandemic altered consumers' grocery buying habits (**Figs 5** and **6**). 54% of respondents indicated that the pandemic altered the frequency of their in-store

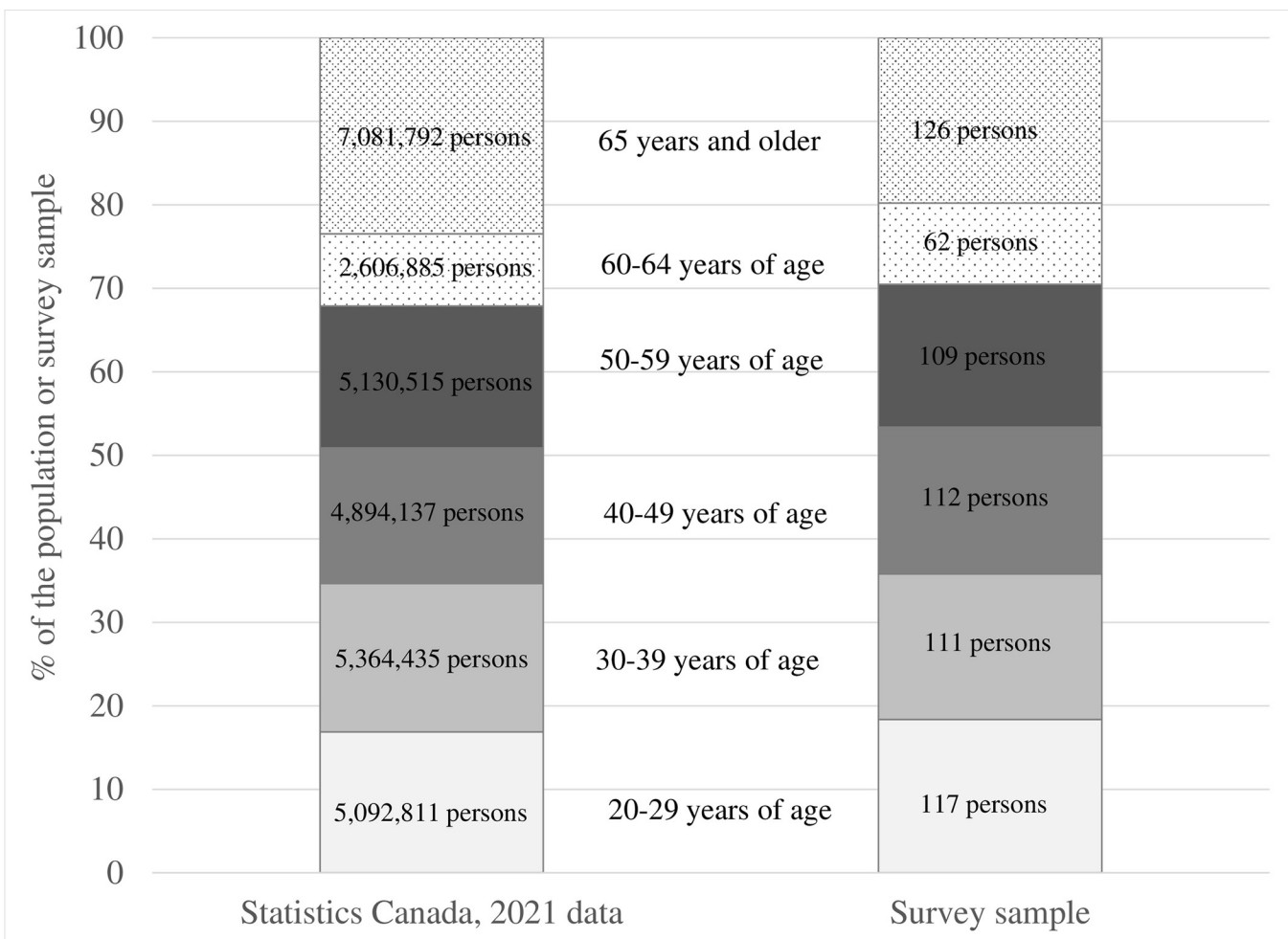

**Fig 4. Distribution by age, study sample versus Statistics Canada data.**

grocery shopping visits. Only 35 of the 651 consumers surveyed returned to the "pre-COVID normal" in-store visits as the pandemic progressed, while 108 respondents indicated that, almost two years into the pandemic, they did not visit grocery stores as frequently as before. Overall, when respondents were asked if the onset of the pandemic changed their grocery shopping behavior considering the frequency of store visits, time spent in stores, or frequency of online grocery purchases, the overwhelming majority (72%) recognized that their behavior changed a lot (264 out of 651 participants) or somewhat changed (207/651) during the first wave of the pandemic. **Fig 7** illustrates the importance of various factors in triggering the change in grocery shopping behavior of Canadian consumers. As one can see, health concerns for self and family members were mentioned as very important by the overwhelming majority of the respondents.

Of those 471 consumers who reported that they altered their grocery shopping behavior one way or another during the first wave of the COVID-19 pandemic, only 164 (35%) reported a return to 'pre-COVID normal" as of January 2022 –almost 2 years into the pandemic. The remaining 307 (65%) of consumers said their shopping behavior had not returned to "pre-COVID normal" as of January 2022. When asked whether they thought their behavior would return to "pre-COVID normal" in the future, only nine respondents answered "definitely yes"

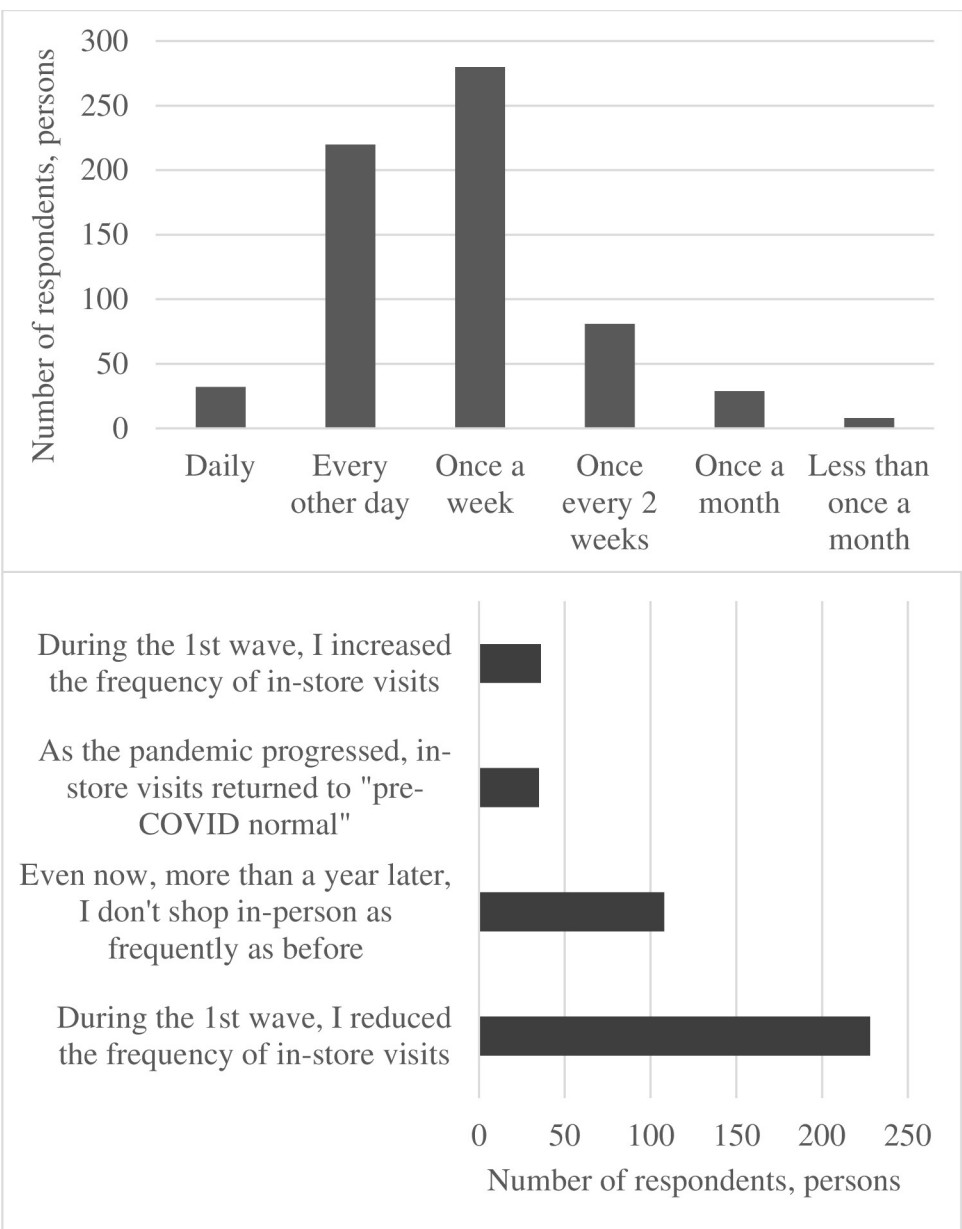

**Fig 5.** Frequency of in-store visits prior to the pandemic (top) and the change that occurred as a result of the pandemic (bottom).

and 204 participants (46% of the full study sample) reported the changes in their behavior that occurred during COVID will persist in the future.

## 5.2 Has the COVID-19 pandemic accelerated adoption of online grocery shopping?

In our study sample, 55% of the respondents (387) associated themselves with 'not online grocery shoppers', 161 participants (25% of the sample) identified themselves as first-time online grocery shoppers, and 129 participants (20%) indicated that they had purchased groceries online prior to the pandemic.

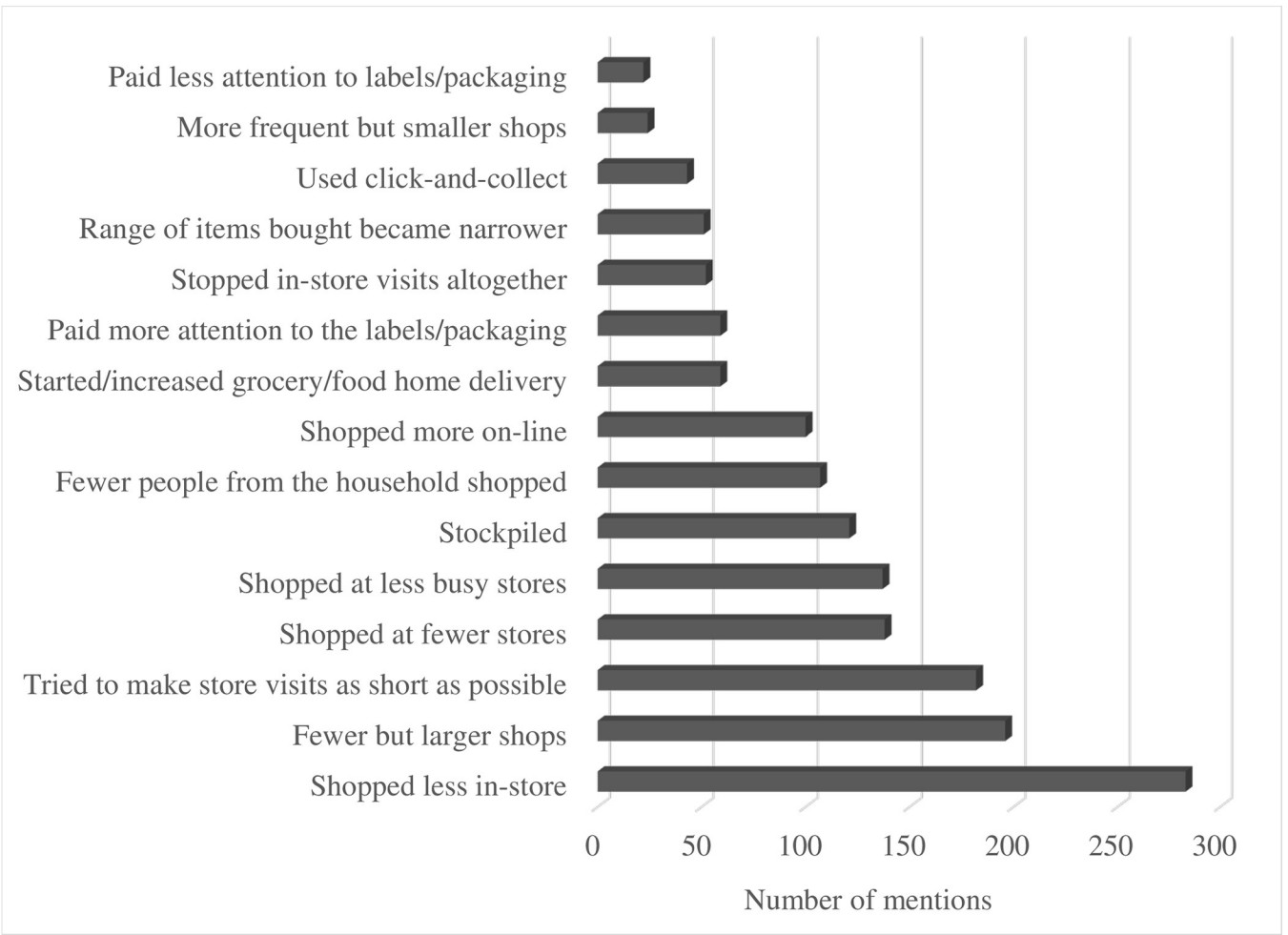

**Fig 6. Major changes that occurred in grocery shopping behavior during the first wave of COVID-19.**

Fig 8 illustrates the various factors that first-time online grocery buyers identified important in shifting some or all of their grocery purchases online. Not surprisingly, the desire to maintain social distance (minimize human contact) for their and their family members' health was one of the most important factors. Despite the fact that the desire to maintain social distance is likely to disappear when the COVID-19 pandemic is over, only 24 of the 161 first-time online shoppers indicated that they will not purchase groceries on-line post pandemic. The top two reasons for discontinuance of online grocery shopping were #1 inability to inspect the product and as a result concerns about product quality and #2 the [long] time it takes for an online grocery order to be delivered or be ready for pickup.

With respect to the on-going online grocery shoppers, 57% (74 out of 129 participants) reported an increase in their online grocery purchases during the pandemic. The various reasons for increased purchases followed a very similar pattern to what we observed for the first-time online shoppers. The survey results reveal that 82% of the ongoing online shoppers who increased their online purchases during the pandemic, expect to maintain the same behavior (i.e. buy more groceries online) post pandemic. Again, this result reinforces our finding for the first-time online grocery shoppers that most consumers will not revert back to their "pre-COVID normal".

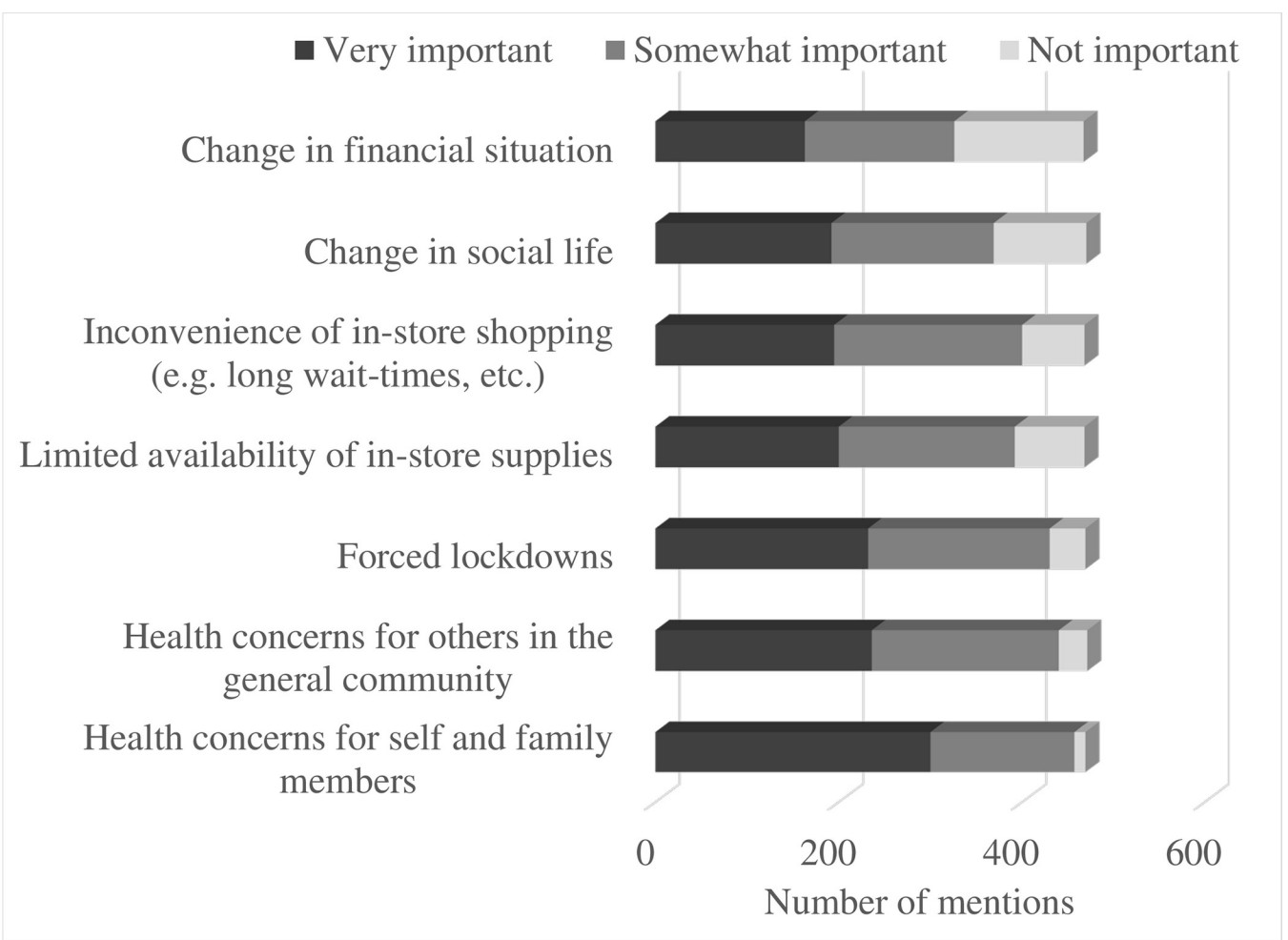

**Fig 7. The importance of various factors in triggering the change in grocery shopping behavior.**

**Table 2** below provides an insight into the role of some key consumer characteristics in explaining demand for online grocery shopping prior to the pandemic and in creating new demand during the pandemic. As one can see from the table, for certain consumer types, the new demand—the number of respondents who started online grocery shopping during the pandemic—was almost twice as high as the demand prior to the pandemic. Consumer type captures differences in perceived risk and fear of the virus. More specifically, in the survey questionnaire, we asked respondents to identify themselves with one of the following consumer types, depending on how they felt during the pandemic: (1) Worrier/Concerned–"I am very fearful of the future and I worry about my health a lot. I'm not willing to take any chances. News about new COVID cases and deaths caused a fair amount of stress" [28% of the sample]; (2) Individualist–"I and my family will be fine. I am more concerned about people acting irrational and engaging in unreasonable panic buying" [15% of the sample]; (3) Rationalist–"I am not concerned. All I can do is keep things and myself clean. I hope others do the same. New doesn't worry me too much as I have this 'keep calm and carry on' mentality" [20% of the sample]; (4) Activist–"I want to maintain social distance and wear masks not just to protect myself but to protect others as well. Protecting others is our social responsibility" [28% of the sample]; (5) Indifferent–"This is seriously being blown out of proportion; it's just a flu. All these

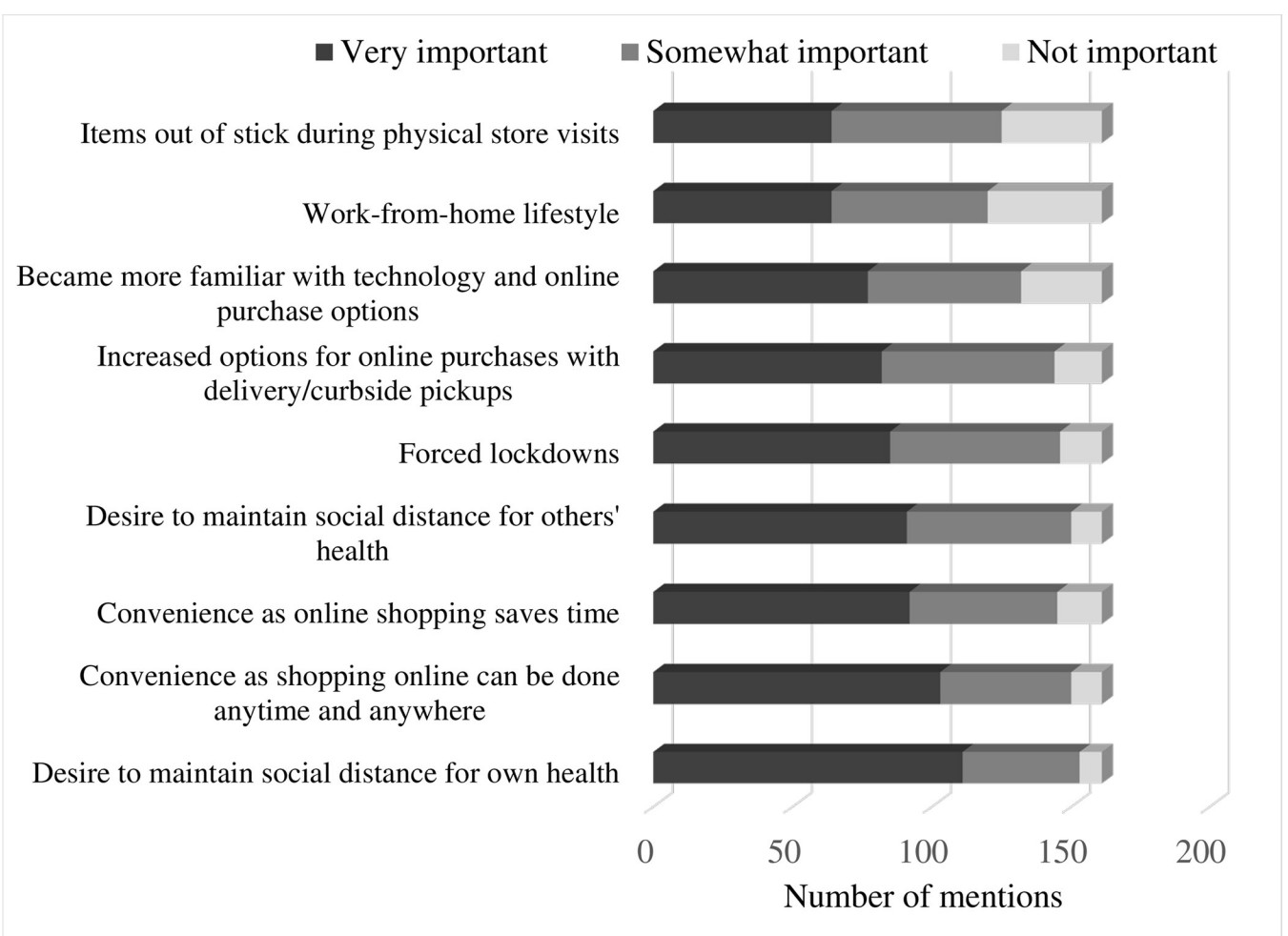

**Fig 8. Reasons for starting online grocery purchases.**

government rules and restrictions to confine the spread of the virus make no sense and only create inconveniences" [9% of the sample]. Based on the definitions, one could hypothesize that "Worriers" and "Activists" would be more likely to take measures to avoid or limit in-person grocery shopping due to elevated fears for their own health (worriers) or concerns about the health of others in the society (activists). As **Table 2** indicates, new demand for online grocery shopping coming from these two consumer types appears much larger than new demand from the other three types (individualist, rationalist, and indifferent), with over one third of the respondents who identified themselves as "worriers" or "activists" becoming first-time online grocery shoppers during the pandemic. In contrast, the pandemic triggered the adoption of online mode of grocery shopping for only about 10% of the "Indifferent".

In the existing literature, the role of gender in adoption of online grocery shopping is ambiguous. The data from our sample illustrate that the percent of female consumers who bought groceries online prior to the pandemic is very close to that of the male consumers (**Table 2**). In terms of adoption of online mode of grocery shopping, the responses of the two groups to the COVID pandemic also seem similar, with about 27% of the female consumers reporting that they started purchasing groceries online as a result of the pandemic versus 22% for men.

**Table 2. New demand for online grocery shopping and key consumer characteristics.**

| Key consumer characteristics | Number of consumers with the said characteristic | Existing pre-COVID demand: Bought groceries online prior to COVID | New demand: Started buying groceries online due to COVID | Part of the new demand to be sustained in the future |
|---|---|---|---|---|
| *Consumer type* | | | | |
| Worrier | 181 | 38 (21.0%[1]) | 60 (33.1%[2]) | 41 (68% of 60) |
| Individualist | 98 | 24 (24.5%) | 19 (19.4%) | 14 (74% of 19) |
| Rationalist | 129 | 23 (17.8%) | 16 (12.4%) | 9 (56% of 16) |
| Activist | 186 | 30 (16.1%) | 60 (32.3%) | 42 (70% of 60) |
| Indifferent | 57 | 14 (24.6%) | 6 (10.5%) | 3 (50% of 6) |
| *Gender* | | | | |
| Males | 272 | 49 (18.0%) | 59 (21.7%) | 37 (63% of 59) |
| Females | 375 | 77 (20.5%) | 101 (26.9%) | 71 (70% of 101) |
| *Presence of children* | | | | |
| Households without children | 487 | 91 (18.7%) | 110 (22.6%) | 73 (66% of 110) |
| Households with children | 164 | 38 (23.2%) | 51 (31.1%) | 36 (71% of 51) |
| • With children under 6 | 59 | 13 (22.0%) | 23 (39.0%) | 16 (70% of 23) |
| • With children 6–11 | 56 | 14 (25.0%) | 19 (33.9%) | 14 (74% of 19) |
| *Age* | | | | |
| • Younger than 40 | 242 | 68 (28.1%) | 77 (31.8%) | 57 (74% of 77) |
| • 40–59 years old | 221 | 41 (18.6%) | 55 (24.9%) | 34 (62% of 55) |
| • 60 and older | 188 | 20 (10.6%) | 29 (15.4%) | 18 (62% of 29) |
| *Educational attainment* | | | | |
| • High school and less | 232 | 53 (22.8%) | 35 (15.1%) | 24 (69% of 35) |
| • Certificate/diploma | 254 | 45 (17.7%) | 65 (25.6%) | 42 (65% of 65) |
| • University degree | 165 | 31 (18.8%) | 61 (37.0%) | 43 (70% of 61) |
| *Annual household income* | | | | |
| • Less than $20,000 | 171 | 32 (18.7%) | 46 (26.9%) | 34 (74% of 46) |
| • $20,000 - $59,999 | 294 | 66 (22.4%) | 67 (22.8%) | 42 (63% of 67) |
| • $60,000 - $99,999 | 128 | 26 (20.3%) | 30 (23.4%) | 21 (70% of 30) |
| • $100,000 and above | 57 | 5 (8.8%) | 18 (31.6%) | 12 (67% of 18) |
| *Province of residence* | | | | |
| • NFL (4,348[3]) | 43 | 4 (9.3%) | 15 (34.9%) | 11 (73% of 15) |
| • PEI (1,078) | 23 | 4 (17.4%) | 5 (21.7%) | 3 (60% of 5) |
| • NS (8,132) | 45 | 6 (13.3%) | 15 (33.3%) | 8 (53% of 15) |
| • NB (6,115) | 43 | 5 (11.6%) | 9 (20.9%) | 6 (67% of 9) |
| • QB (61,023) | 92 | 24 (26.1%) | 20 (21.7%) | 17 (85% of 20) |
| • ON (94,628) | 124 | 29 (23.4%) | 40 (32.3%) | 25 (63% of 40) |
| • MB (9,739) | 67 | 18 (26.9%) | 16 (23.9%) | 8 (50% of 16) |
| • SK (8,408) | 66 | 9 (13.6%) | 12 (18.2%) | 7 (58% of 12) |
| • AB (24,112) | 73 | 12 (16.4%) | 18 (24.7%) | 16 (89% of 18) |

*(Continued)*

**Table 2.** (Continued)

| Key consumer characteristics | Number of consumers with the said characteristic | Existing pre-COVID demand: Bought groceries online prior to COVID | New demand: Started buying groceries online due to COVID | Part of the new demand to be sustained in the future |
|---|---|---|---|---|
| • BC (33,903) | 75 | 18 (24.0%) | 11 (14.7%) | 8 (72% of 11) |

[1] –In brackets, we report the percent of consumers in the respective consumer group who bought groceries on-line prior to the pandemic.

[2] –In brackets, we report the percent of consumers in the respective consumer group who started buying groceries on-line as a result of the pandemic.

[3] –In brackets, we report COVID-related total number of deaths from March 3, 2020 to December 31, 2020. Data retrieved from Statistics Canada, CANSIM, Table 13-10-0784-01.

The data in **Table 2** also illustrate that new demand for online grocery shopping for households with children was higher than that for households without children. Almost 40% of households with children under the age of 6 reported that they had started buying groceries online when the pandemic hit. In contrast, only about one in five households without children responded to the COVID-19 pandemic by adopting online grocery shopping.

**Table 2** shows that younger consumers (under the age of 40) were more likely to buy groceries online prior to the pandemic. Although the percent of first-time online grocery shoppers appears higher for younger consumers, the extent of the shift in demand, relative to the existing pre-COVID demand, seems to be larger for older consumers.

The data provides some evidence that the rate of adoption of online mode of grocery shopping was higher for more educated consumers, although the share of consumers buying groceries online prior to the pandemic was slightly higher for the least educated consumer group (high school and less). While the highest income consumer category ($100,000 and above) saw the smallest share of on-line grocery shoppers prior to the pandemic, this category experienced the largest increase (shift) in demand for online grocery purchases as a result of the pandemic.

The survey results suggest (**Table 2**, last column) that many consumers will not discontinue the online mode of grocery shopping whenever the initial trigger disappears, which provides some evidence to suggest that the process of diffusion of the online grocery innovation follows the smooth and continuous path suggested by some traditional definitions of adoption. While this finding is consistent with the results for online grocery market in Germany [59, 60], it is in contrast to the findings of [61] who find that many consumers discontinue the online mode of shopping whenever the initial triggers disappear and contrary to the arguments posed by [108]. The contrast in the findings can potentially be explained by the duration of the triggers. When the initial triggers such as illness that disables an individual from visiting a grocery store in person do not last long, then it is very likely that the shopping behavior will return to the pre-trigger normal. However, our results suggest that for the triggers of longer duration like the COVID-19 pandemic that lasted for more than a year, it is very likely that the changes in consumer behavior will grow into new habits that will prevail in the future. However, the fact that a small portion of consumers will return to "brick-and-mortar" grocery shopping also tells us that the adoption decision seems to be re-evaluated frequently. As such, post-adoption evaluation appears crucial to the decision of whether to continue with the innovation. When one looks at online grocery shopping adoption, some degree of innovation adoption discontinuity occurs because the adoption of online shopping is complementary to buying in stores, rather than substitutive. Reverting back to the traditional mode of shopping is easy because most consumers never completely cease to shop in traditional stores.

An interesting observation is that out of 235 consumers who started or increased their online grocery shopping activities during the pandemic, only 116 (49%) participants reported a reduction in the frequency of their in-store grocery shopping visits. This suggests that the

observed increase in online grocery shopping throughout the pandemic was not motivated solely by the desire to limit contact with others through substitution of some of the in-store purchases with online; consumers could potentially be motivated to start purchasing groceries online because the stores expanded options for online orders and order pickups.

While **Table 2** reports useful information about the distribution of consumers in our sample in terms of some key consumer characteristics as well as provides an initial insight into what factors could be important in explaining differences in consumer response to the COVID pandemic in terms of adoption of online grocery shopping, it does not allow us to make inferences if these differences are statistically significant. **Table 3** below reports the results of the pairwise tests analysis, showing Z-scores for differences in proportions. The differences are defined in such a way so that the null hypothesis that is being tested is consistent with the propositions developed in section 3 above.

One important observation from **Table 3** is that none of the consumer characteristics except age seems to be important in explaining differences in pre-COVID demand for on-line grocery shopping. While some Z-scores are statistically significant when one analyzes the differences across households with different income levels, there is no consistency in the results that would suggest that the proportion of online grocery shoppers prior to COVID was higher for higher income households. The only income group that stands out in terms of pre-COVID demand is $100,000+ group. So, the results reveal that before the pandemic the only factor truly explaining differences in adoption of online grocery shopping was age: younger consumers were more likely to purchase groceries online.

In Proposition 1a we hypothesized that a province of residence can be a proxy for consumer information about COVID; given the varying number of COVID-related deaths across the provinces, consumers living in provinces with higher deaths count are hypothesized to have higher perceived fears. {QB, ON} are the two provinces with the highest deaths count between March 2020 and December 31, 2020; {AB; BC} rank the second highest. All other provinces reported much lower COVID-related deaths numbers than {QB; ON} and {AB; BC}. The results show that the proportion of consumers who started using online grocery shopping due to COVID in {AB; BC} is not statistically different than the proportion of first-time online shoppers in the provinces with much lower reported number of deaths (SK, MB, NB, NS). So, from this pairwise hypothesis testing we do not find evidence to support Proposition 1a.

In Proposition 1b we hypothesized that consumers with more negative style of thinking (worriers in nature) are more likely to have higher perceived fears and, as a result, more likely to undertake measures to limit their exposure to the virus, including switching their grocery purchases to online. By comparing "Worriers" to "Individualists", "Rationalists", and "Indifferent" we find that the proportion of "Worriers" who started using online model of grocery shopping during COVID is significantly higher (at 1% significance level) than the proportions for these three other types of consumers that by definition had much lower concerns about their own health or the health of others in the society. Note that for pre-COVID demand, there were no statistically significant differences in proportions; significant differences in proportions for new demand, however, suggest that differences in perceived risks and fears–a factor specific to pandemics—is an important trigger of adoption of online grocery shopping.

In proposition 2 we hypothesized that socio-economic characteristics of a household played an important role in developing response to the COVID pandemic. More specifically, in Proposition 2c we suggested that the presence of small children in a household could contribute to elevated perceived fears as small children may be more likely to develop complications from viral infections due to underdeveloped immune systems. As the results in **Table 3** indicate we do find evidence that, compared to households without children, a significantly higher proportion of households with children started purchasing groceries online during the pandemic (the

**Table 3. Pairwise tests analysis: Difference in proportions.**

| Key consumer characteristics | Pre-COVID demand for online grocery shopping | | New demand for online grocery shopping (first-time buyers) | | Evidence to support: |
|---|---|---|---|---|---|
| | Difference in proportions (standard error) | Z-score $H_1: p_{difference} \neq 0$ | Difference in proportions (standard error) | Z-score $H_1: p_{difference} > 0$ | |
| *Province of residence ($p_{difference} = p_{higher\ COVID\ death\ count} - p_{lower\ COVID\ death\ count}$)* | | | | | |
| {ON, QB} vs {AB, BC} | 0.042 (0.044) | 0.938 | 0.082 (0.045) | 1.788** | No evidence to support Proposition 1a |
| {AB, BC} vs {SK, MB, NB, NS} | 0.031 (0.042) | 0.753 | -0.039 (0.043) | -0.887 | |
| *Consumer type ($p_{difference} = p_{worrier} - p_j$)* | | | | | |
| Worrier vs Individualist | -0.035 (0.053) | -0.671 | 0.138 (0.053) | 2.435*** | Proposition 1b |
| Worrier vs Rationalist | 0.032 (0.045) | 0.697 | 0.208 (0.045) | 4.186*** | |
| Worrier vs Activist | 0.049 (0.041) | 1.206 | 0.009 (0.049) | 0.174 | |
| Worrier vs Indifferent | -0.036 (0.065) | -0.575 | 0.227 (0.054) | 3.331*** | |
| *Presence of children ($p_{difference} = p_{with} - p_{without}$)* | | | | | |
| Households w/children vs households w/o children | 0.045 (0.037) | 1.250 | 0.085 (0.041) | 2.182** | Proposition 2c |
| Households w/ children under 6 vs w/children 6–11 | -0.030 (0.079) | -0.380 | 0.051 (0.090) | 0.568 | |
| *Age ($p_{difference} = p_{younger} - p_{older}$)* | | | | | |
| Less than 40 vs 40–59 | 0.095 (0.039) | 2.406** | 0.069 (0.042) | 1.643* | Proposition 3 |
| Less than 40 vs 60+ | 0.175 (0.037) | 4.463*** | 0.164 (0.039) | 3.915*** | |
| 40–59 vs 60+ | 0.080 (0.034) | 2.263** | 0.095 (0.039) | 2.370*** | |
| *Educational attainment ($p_{difference} = p_{more\ education} - p_{less\ education}$)* | | | | | |
| Certificate/diploma vs high school or less | -0.051 (0.036) | -1.400 | 0.105 (0.036) | 2.860*** | Proposition 4 |
| University degree vs Certificate diploma | 0.011 (0.039) | 0.286 | 0.114 (0.047) | 2.486*** | |
| University degree vs high school or less | -0.040 (0.041) | -0.962 | 0.219 (0.044) | 5.021*** | |
| *Annual household income ($p_{difference} = p_{higher\ income} - p_{lower\ income}$)* | | | | | |
| $20,000-$59,999 vs less than $20,000 | 0.037 (0.038) | 0.944 | -0.041 (0.042) | -0.994 | No evidence to support Proposition 2a and Proposition 5 |
| $60,000-$99,999 vs less than $20,000 | 0.016 (0.046) | 0.346 | -0.035 (0.051) | -0.688 | |
| $100,000+ vs less than $20,000 | -0.099 (0.048) | -1.756* | 0.047 (0.070) | 0.684 | |
| $60,000-$99,999 vs $20,000-$59,999 | -0.021 (0.043) | -0.481 | 0.006 (0.045) | 0.135 | |
| $100,000+ vs $20,000-$59,999 | -0.136 (0.045) | -2.341** | 0.088 (0.066) | 1.419* | |
| $100,000+ vs $60,000 - $99,999 | -0.115 (0.052) | -1.934* | 0.082 (0.072) | 1.175 | |
| *Gender ($p_{difference} = p_{females} - p_{males}$)* | | | | | |
| Females vs Males | 0.025 (0.031) | 0.793 | 0.052 (0.034) | 1.514* | Proposition 7 |

Note

*** Significant at 1%

** Significant at 5%

* Significant at 10%

For pre-COVID demand, we are testing a two-tailed hypothesis $H_0: p_{difference} = 0$ vs $H_1: p_{difference} \neq 0$.

For new demand due to COVID, we are testing a left-tail hypothesis where $H_0: p_{difference} \leq 0$ vs $H_1: p_{difference} > 0$

difference is significant at 5%). Our results, however, do not support a premise that households with children under the age of 6 were more likely to adopt online grocery shopping than households with older children—children aged 6 to 11.

There are statistically significant differences in the estimated proportions of younger consumers who started using online grocery shopping during the pandemic versus older

consumers. The difference between the "less than 40" consumer group and "40–59" consumer group is significant at 10%, while the differences between "less than 40" and "60+" consumers and between "40–59" and "60+" consumers are both significant at 1%. The hypothesis testing result therefore provides evidence to support Proposition 3.

The results in **Table 3** also reveal that the estimated proportions of online grocery shopping adopters are significantly higher for consumers with higher education level (Proposition 4; all differences are significant at 1%), while we find no evidence to support our hypothesis that household income is a significant predictor of adoption of online grocery shopping.

While hypothesis testing results reported in **Table 3** are useful in identifying the factors explaining why some consumers chose to switch some of their grocery purchases to the online channel, hypothesis testing does not take into account the association between the study variables. For example, households with small children are likely to also fall in the "younger" consumer group and when we find that the proportion of households with small children that adopted online grocery shopping is significantly higher than that for households without dependent children, the association between the presence of small children and age may lead to the conclusions that do not reflect the importance of a household having small children in triggering change in their grocery shopping habits. Multiple regression analysis provides a way of accounting for potentially confounding variables, thus separating the contribution of each separate factor to the change in the outcome variable–increased use or adoption of online grocery shopping.

In the following, we present the results of the econometric analysis to better understand the importance of different factors in stimulating consumers to switch their grocery purchases online.

## 5.3 The results of econometric estimation and discussion

**Table 4** shows the estimated marginal effects for the Logit models (see **S1 File** for detailed variable description). The estimated effects for the most part are consistent with the previous research findings and with our expectations.

The results reveal that personality type played a role in the consumer's decision to adopt online grocery shopping. More specifically, the "worriers/concerned" were 19% more likely to switch to online grocery purchases than the benchmark. The benchmark group included "indifferent", "individualist", and "rationalist" that by definition were not concerned about COVID itself but were more concerned about irrational behavior of other consumers and inconveniences caused by the imposed government restrictions. The "activists" are found to be 21% more likely to switch to online grocery purchases.

**5.3.1 Factors that determine the adoption or increased use of online grocery shopping during the pandemic (Model 1).** *COVID-related fears.* COVID-related fears are found to have significantly contributed to adoption of online grocery shopping during the pandemic, which is in support of Proposition 1. Interestingly, it is perceived susceptibility (fears associated with catching the virus) rather than perceived severity (fears associated with developing health complications in case of illness) that played a significant role in consumers' decisions to switch all or some of their grocery purchases online: those who were very concerned and those who were somewhat concerned about contracting the virus are found to be 20% and 23%, respectively, more likely to switch to online grocery purchasing format; both estimates are statistically significant at 1% significance level. The results also show that the presence of severe COVID cases among the consumer's contacts (family members, colleagues, or friends)–a proxy for knowledge about COVID gained from personal experience—also had a significant impact on the decision to purchase more groceries online.

**Table 4. Factors that explain adoption of online grocery shopping: Logit estimation results.**

| Explanatory variables | Model 1: Adoption /increased use of online grocery shopping during the pandemic | | Model 2: Intention to use online grocery shopping channel more in the future | |
|---|---|---|---|---|
| | Marginal effect (prob($Y_1$)) | Robust standard error | Marginal effect (prob($Y_2$)) | Robust standard error |
| Worrier | 0.189*** | 0.060 | - | - |
| Activist | 0.211*** | 0.058 | - | - |
| Perceived susceptibility (high) | 0.200*** | 0.078 | - | - |
| Perceived susceptibility (somewhat high) | 0.226*** | 0.059 | - | - |
| Perceived severity (high) | 0.053 | 0.084 | - | - |
| Perceived severity (somewhat high) | 0.011 | 0.074 | - | - |
| COVID_close_contacts | 0.109** | 0.053 | - | - |
| QB | -0.021 | 0.069 | - | - |
| ON | 0.044 | 0.067 | - | - |
| MB | 0.004 | 0.079 | - | - |
| SK | -0.091 | 0.068 | - | - |
| AB | -0.077 | 0.064 | - | - |
| BC | -0.083 | 0.065 | - | - |
| Children_ages_6–11 | -0.009 | 0.074 | 0.010 | 0.098 |
| Children_under 6 | 0.107 | 0.015 | 0.126 | 0.095 |
| Income | -0.027* | 0.015 | -0.038 | 0.026 |
| Change in income (reduction) | 0.113** | 0.045 | - | - |
| Change in income (increase) | 0.021 | 0.095 | - | - |
| Age | -0.004*** | 0.001 | -0.001 | 0.002 |
| Education | 0.073*** | 0.017 | 0.087*** | 0.029 |
| Sex (female) | 0.055 | 0.043 | 0.064 | 0.075 |
| Prior frequency of store visits | 0.035** | 0.016 | 0.037 | 0.038 |
| Shopping venue (large supermarkets) | -0.013 | 0.059 | -0.033 | 0.088 |
| Prior online | 0.005*** | 0.001 | - | - |
| Grocery_importance | -0.002** | 0.001 | -0.003* | 0.001 |
| **Past experience with online grocery shopping** | | | | |
| Past experience | - | - | 0.150* | 0.078 |
| **Perceived advantages** | | | | |
| Time savings | - | - | 0.029 | 0.042 |
| Economical | - | - | 0.006 | 0.042 |
| Greater product variety | - | - | -0.022 | 0.034 |
| Ease of use | - | - | 0.026 | 0.043 |
| Pleasure | - | - | 0.122*** | 0.043 |
| Benefits | | | 0.101*** | 0.039 |
| **Perceived disadvantages** | | | | |
| Over-spending | - | - | 0.005 | 0.031 |
| More costly | - | - | -0.017 | 0.039 |
| Quality of fresh produce | - | - | 0.028 | 0.035 |
| Pseudo R-square | 0.2120 | | 0.1814 | |
| Wald chi2(29) [model significance] | 119.75*** | | 30.13*** | |
| Number of observations | 650 | | 288 | |

Note

*** Significant at 1%

** Significant at 5%

* Significant at 10%

We don't find a statistically significant difference in adoption of online grocery shopping based on the province of residence, which was hypothesized to serve as a proxy for consumer information about the spread of COVID and COVID-related deaths.

*Socio-economic background.* Among the variables that capture socio-economic background of the respondents, only the income and a reduction in income due to the pandemic are found to be statistically significant at 10% and 5%, respectively. More specifically, consumers with higher incomes were less likely to switch more of their grocery purchases online, while consumers whose incomes decreased as a result of the pandemic were more likely to do so. This finding is in contrast to the hypothesized relationship (Propositions 2a, 2b, Proposition 5) based on the TAM (perceived ease-of-use) framework; however, it can make perfect sense if one thinks about the effect of income on online purchases in the context of the valuation of time: higher-income consumers value their time more because of its opportunity cost [116]. [117] notes that "consumers who are "time rich and income poor" find online shopping to be attractive mainly for the money savings potential, while those who are "income rich and time poor" may be attracted to it because it saves time". During the pandemic, a reduction in income was associated with temporary/permanent layoffs or reduction in working hours, which would make the affected consumers time rich and free up more time for online shopping in general and for groceries in particular. Also, those consumers whose incomes were negatively affected by the pandemic might have faced higher emotional stress. Studies have found that higher levels of stress may impact eating behaviors, especially eating more unhealthy foods such as sugary, high-fat, and savory foods [118]; this group of foods is very common in online grocery purchases because unlike fresh produce consumers are usually not as concerned about quality not meeting expectations. Therefore, our findings suggest that most likely it is the opportunity cost of time of online shopping that played a role in creating incentives for consumers to switch to online grocery shopping rather than technological abilities and opportunities postulated to be captured by the income variable. Indeed, when asked about the reasons not to shop for groceries online, the non-adopters placed "problems with technology use and/or internet access" at the bottom of the list in terms of importance. This finding and the Pearson chi-square test for association between income and importance of the "problems with technology use and internet access" reason for non-adopters (p-value of 0.49), suggest that for non-adopters technological constraints to perform online shopping are minimal and are independent of the income level.

Proposition 2c that the presence of small children would significantly impact the likelihood of switching online is not supported by the multiple regression results.

*Pre-COVID grocery shopping behavior.* The results support our hypothesis (Proposition 6) that pre-COVID shopping behavior played an important role in explaining why some consumers switched to online and some didn't. Those consumers who had more frequent grocery store visits prior to the pandemic are found to be more likely to either have started or increased their online grocery purchases during the pandemic, which could be the result of some substitution between in-store and online grocery purchases. The results reveal that the share of grocery store purchases in average weekly meals is an important predictor whether a consumer switched to online grocery shopping during the pandemic. Based on our survey sample, slightly less than 70% of an average consumer's weekly meals were meals cooked at home from scratch from ingredients bought from grocery stores and slightly under 10% of all meals were ready-to-eat items or heat-and-eat meals from grocery stores. So, an average consumer would require items from grocery stores for almost 80% of their meals. The results indicate that someone who requires grocery store items for 90% of their meals (i.e. a 10% increase from an average consumer) is 2% less likely to have adopted or increased online grocery shopping during the pandemic. The estimation results also reveal that those consumers who had a largest

portion of their grocery purchase done online prior to COVID were more likely to increase their online grocery purchases during the pandemic.

*Age and gender*. We find that age is a significant predictor of online grocery shopping adoption. The role of age in this model is two-fold. On the one-hand, higher age can be a COVID risk factor, thus creating more incentives for older consumers to alter their grocery buying behavior, including a reduction in frequency of in-store visits and a switch to online grocery shopping to replace some of in-store purchases. Yet, the results of a bi-serial correlation do not support any significant association between age and a reduced frequency of in-store visits (p-value 0.1321). On the other hand, age can be related to technological abilities. We find that age is significantly and negatively associated with the probability of increasing online grocery purchases during the pandemic: for every 10 years increase in age, the probability of adopting or increasing online grocery purchases is reduced by 4%. This result supports Proposition 3 and suggests that age variable is more important to capture technological opportunities and abilities than the COVID-related risks and fears.

We do not find any evidence that either gender is more psychologically predisposed to adoption of online grocery shopping.

*Education*. The results indicate that education plays a significant role in explaining consumers' decisions to adopt online grocery shopping: higher level of educational attainment is associated with a higher probability of technology adoption, which is consistent with the TAM (Proposition 4).

**5.3.2 Factors that determine the use of online grocery shopping in the post-pandemic world (Model 2).** We find that the significant determinants of sustaining the "new" demand for online grocery shopping in the future include education, positive past experience, two of the six perceived advantages of online grocery shopping, and importance of grocery store purchased items in family meals.

*Education*. Consistent with the TAM and TCT postulating that more educated individuals are more likely to adopt new technologies and continue using them in the future, our findings suggest that consumers with higher educational attainment levels are more likely to continue the behavior that emerged during the pandemic.

*Past experience*. The results indicate that past experience is a significant predictor of future behavior. Those consumers who reported that they had positive experience with their previous online purchases are 15% more likely to use online modes for their future grocery purchases compared to those who had neutral or negative experiences. Furthermore, those consumers who reported a higher level of agreement with the following statements "I view online grocery shopping as a pleasure rather than a chore" and "While I started online grocery shopping as a social distancing measure during COVID, I can now see lots of other benefits of shopping for groceries online" are found to be more likely to sustain the trend in online shopping behavior that arose out of the pandemic.

*Perceived advantages*. The survey results also indicate, that once the consumers are triggered to utilize online grocery shopping, they seem to realize there are many benefits associated with it (e.g. time saving–see **Fig 9**).

**Table 5** reports the average scores for the disadvantages of online grocery shopping. The average scores and the results of the paired t-tests suggest that of the three suggested disadvantages, quality of fresh produce is by far the biggest challenge with online grocery shopping. Indeed, consumers tend to purchase non-perishable items online. Among those consumers who switched some of their purchases to online models as a result of the pandemic, 95% said they purchased non-perishable items such as pasta, flour, and cereal online, while only 84% and 79% said they purchased fresh produce and fresh meat/fish, respectively. Therefore, fresh meat/fish seems to be the least popular food category for online grocery purchases.

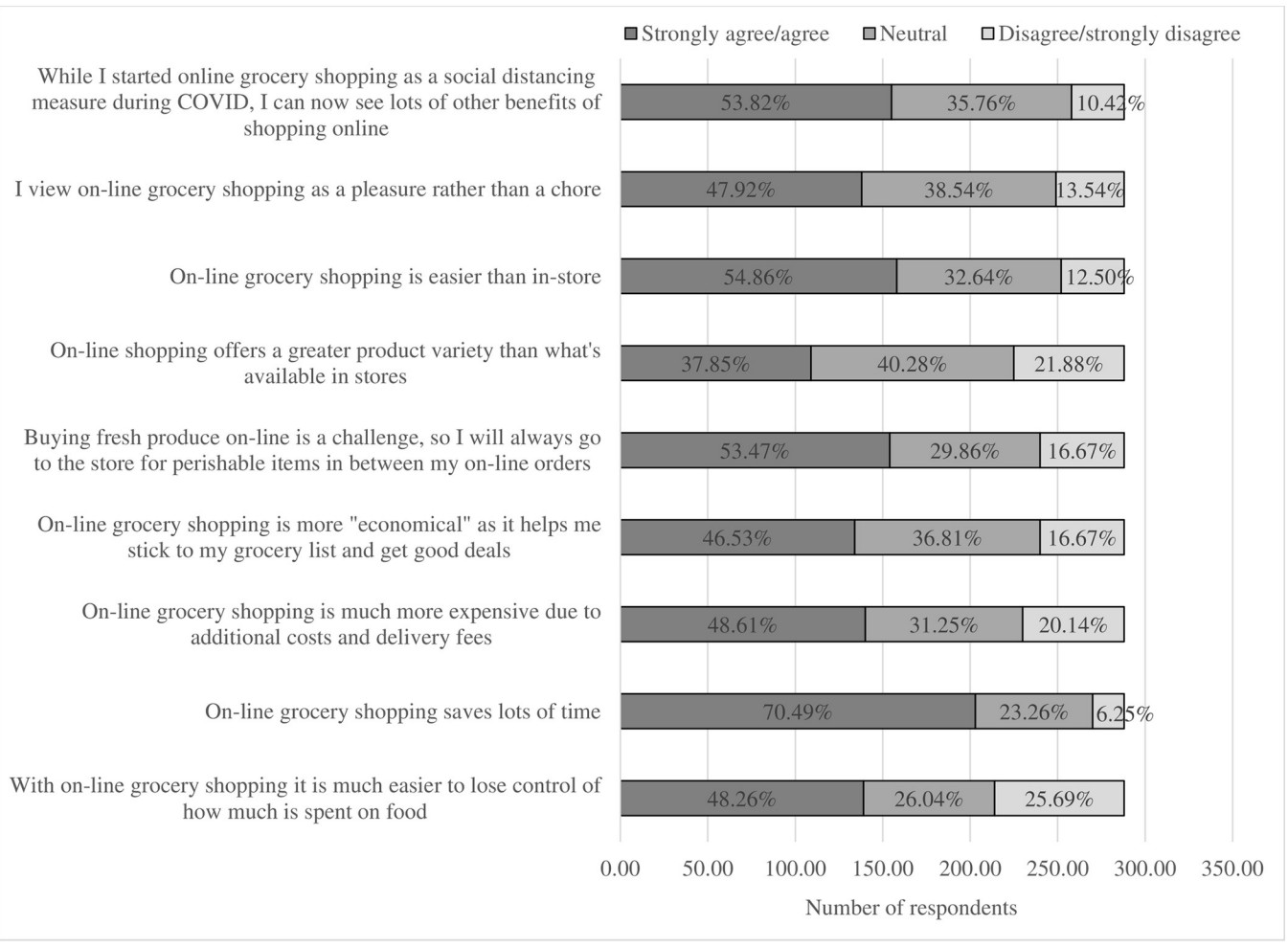

**Fig 9. Assessment of some potential advantages and disadvantages of online grocery shopping.**

**Table 5. Consumers' perception of disadvantages of online grocery shopping.**

| Disadvantages of online grocery shopping | Average score[1] | Paired t-test [t-stat (p-value)] | |
|---|---|---|---|
| | | Disadv1 | Disadv2 |
| Buying fresh produce is a challenge, so I will always go to the store for perishable items in between my online orders (Disadv1) | 3.54 | - | - |
| Online grocery shopping is much more expensive due to additional costs and delivery fees (Disadv2) | 3.38 | $H_0 : \bar{x}_1 - \bar{x}_2 = 0$ $H_1 : \bar{x}_1 - \bar{x}_2 > 0$ 2.05** (0.0204) | - |
| With online grocery shopping it is much easier to lose control of how much is spent on food (Disadv3) | 3.29 | $H_0 : \bar{x}_1 - \bar{x}_3 = 0$ $H_1 : \bar{x}_1 - \bar{x}_3 > 0$ 2.82*** (0.0026) | $H_0 : \bar{x}_2 - \bar{x}_3 = 0$ $H_1 : \bar{x}_2 - \bar{x}_3 > 0$ 1.23 (0.1092) |

[1] Note: the responses are measured on a 5-point Likert scale: 1 strongly disagree– 5 strongly agree; a higher average score indicates a higher level of agreement with the statement.

*** the difference is significant at 1%

** the difference is significant at 5%

**Table 6. Consumers' perception of advantages of online grocery shopping.**

| Advantages of online grocery shopping | Average score[1] | Paired t-test [t-stat (p-value)] | | | | |
|---|---|---|---|---|---|---|
| | | Adv1 | Adv2 | Adv3 | Adv4 | Adv5 |
| Online grocery shopping saves lots of time (Adv1) | 3.98 | - | - | - | - | - |
| Online grocery shopping is easier than in store (Adv2) | 3.60 | $H_0 : \bar{x}_1 - \bar{x}_2 = 0$ $H_1 : \bar{x}_1 - \bar{x}_2 > 0$ 6.05*** (0.000) | - | - | - | - |
| While I started online grocery shopping due to COVID, I can now see lots of other benefits of online shopping (Adv3) | 3.59 | $H_0 : \bar{x}_1 - \bar{x}_3 = 0$ $H_1 : \bar{x}_1 - \bar{x}_3 > 0$ 5.97*** (0.000) | $H_0 : \bar{x}_2 - \bar{x}_3 = 0$ $H_1 : \bar{x}_2 - \bar{x}_3 > 0$ 0.22 (0.41) | - | - | - |
| I view online grocery shopping as a pleasure rather than a chore (Adv4) | 3.44 | $H_0 : \bar{x}_1 - \bar{x}_4 = 0$ $H_1 : \bar{x}_1 - \bar{x}_4 > 0$ 8.52*** (0.000) | $H_0 : \bar{x}_2 - \bar{x}_4 = 0$ $H_1 : \bar{x}_2 - \bar{x}_4 > 0$ 2.89*** (0.000) | $H_0 : \bar{x}_3 - \bar{x}_4 = 0$ $H_1 : \bar{x}_3 - \bar{x}_4 > 0$ 2.40*** (0.009) | - | - |
| Online grocery shopping is more economical as it helps me stick to my grocery list and get good deals (Adv5) | 3.39 | $H_0 : \bar{x}_1 - \bar{x}_5 = 0$ $H_1 : \bar{x}_1 - \bar{x}_5 > 0$ 8.28*** (0.000) | $H_0 : \bar{x}_2 - \bar{x}_5 = 0$ $H_1 : \bar{x}_2 - \bar{x}_5 > 0$ 3.35*** (0.000) | $H_0 : \bar{x}_3 - \bar{x}_5 = 0$ $H_1 : \bar{x}_3 - \bar{x}_5 > 0$ 2.98*** (0.002) | $H_0 : \bar{x}_4 - \bar{x}_5 = 0$ $H_1 : \bar{x}_4 - \bar{x}_5 > 0$ 0.84 (0.20) | - |
| Online shopping offers a greater product variety than what's available in store (Adv6) | 3.24 | $H_0 : \bar{x}_1 - \bar{x}_6 = 0$ $H_1 : \bar{x}_1 - \bar{x}_6 > 0$ 9.75*** (0.000) | $H_0 : \bar{x}_2 - \bar{x}_6 = 0$ $H_1 : \bar{x}_2 - \bar{x}_6 > 0$ 5.28*** (0.000) | $H_0 : \bar{x}_3 - \bar{x}_6 = 0$ $H_1 : \bar{x}_3 - \bar{x}_6 > 0$ 4.82*** (0.000) | $H_0 : \bar{x}_4 - \bar{x}_6 = 0$ $H_1 : \bar{x}_4 - \bar{x}_6 > 0$ 2.99*** (0.002) | $H_0 : \bar{x}_5 - \bar{x}_6 = 0$ $H_1 : \bar{x}_5 - \bar{x}_6 > 0$ 2.25** (0.013) |

[1] Note: the responses are measured on a 5-point Likert scale: 1 strongly disagree– 5 strongly agree; a higher average score indicates a higher level of agreement with the statement.

*** the difference is significant at 1%

** the difference is significant at 5%

Table 6 ranks the statements that capture the potential advantages of online grocery shopping. Statement "online grocery shopping saves lots of time" has received the highest average score among the participants, indicating a higher level of consumers' agreement with this statement compared to all other statements (Table 6). Using the paired t-test results, one can also see that the average score for this statement is significantly higher than for all other statements. The lowest level of consumer agreement is found for the statement that online shopping offers a greater product variety than what is available in store.

## 6. Conclusions

This article analyzed the impact of the COVID-19 pandemic on grocery shopping behavior of Canadian consumers. The results of the study are based off a survey of 651 Canadian consumers across ten Canadian provinces. Our findings suggest that only a small proportion of consumers who altered their shopping habits during the pandemic will revert back to their pre-COVID behavior: 63% of the consumers whose behavior was altered in one way or another reported that the changes will be permanent.

While this study attempted to highlight the changes in consumer behavior in general, the main focus has been on adoption of online mode of grocery shopping. We find that a number of factors played a significant role in consumers' decisions to adopt or increase online grocery shopping during the pandemic. In particular, personality type and perceived susceptibility to acquiring the illness are found to be important; those consumers who are "worriers" or "activists" by nature and those who had higher perceived COVID fears were much more likely to increase their reliance on online grocery purchases. Other significant predictors of online grocery adoption were pre-COVID shopping habits (frequency of grocery store visits prior to the pandemic and importance of store-bought groceries in overall family meals), certain aspects of

socio-economic background (income and reduction in income due to COVID), and the variables capturing technological opportunities and abilities (age and education).

With respect to whether the changes induced by the pandemic would continue in a post-pandemic world, we find that positive experience with online shopping is a significant predictor of continuance with this mode of grocery shopping. The survey results revealed that although many consumers started purchasing groceries online as a measure of social distancing, in general there is a very high level of agreement among consumers that *post factum* they had realized many other benefits of online grocery shopping.

This study has several important implications for the retail food sector. First, one significant contribution of this study is that it reveals that the changes that we have observed during the COVID-19 pandemic are very unlikely to be transient. The majority of the consumers who were triggered by the pandemic to try online grocery shopping for the first time or were triggered to increase their online purchases from the pre-COVID levels have indicated that they will continue this new behavior. Our findings also suggest that online modes of shopping are complementary to buying in stores rather than substitutive; for certain types of products, for example fresh produce and fresh meats and fish, consumers are more likely to revert back to the traditional brick-and-mortar mode of shopping. The ease of reverting back, the intentions of consumers to continue with online grocery shopping, and the importance of past experience with online grocery shopping in predicting future online purchases indicate that grocery providers should make additional investments to accelerate the expansion of online food delivery services as well as focus on improving consumers' experience with online shops. We argue that adoption of online grocery shopping for many will be a continuous innovation, with their new shopping behavior expanding in the future. Therefore, the retail industry should consider significant investments into improving infrastructure for fulfilment of online orders, including in-store automation technology. However, an extensive cost-benefit analysis of such investments should be conducted to ensure positive results for consumers and efficiency without significantly increasing the delivery/order fulfilment cost to consumers. Our results show that almost 50% of the consumers who use online grocery shopping already find that additional costs and delivery fees make online grocery shopping much more expensive compared to the traditional brick-and-mortar shopping method. Investments that will increase order fulfilment efficiency and at the same time create significant additional costs to consumers may discourage future adoption of online grocery shopping and may, therefore, not be justified.

A number of limitations of this study should be warranted. First, the data for this research came from an original survey that, given the research budget constraints, was conducted at one point in time (January 2022). Consumers were asked to document their shopping behavior prior to, at the onset of, and almost two years into the pandemic. Administering the survey at different points in time, thus yielding a longitudinal dataset, would have provided much richer set of information for the analysis. Second, while the existing models of consumer behavior were utilized to lay out the foundation for the econometrics analysis in this study, this econometrics analysis is rather simple in relation to the complexity of the issue. As such, a structural equation modelling (SEM) approach would be a more effective method to validate the existing models of consumer behavior. The survey questions, however, only had a limited number of scale items, which prevented us from properly measuring latent variables within the HBM, TAM, and TCT frameworks to integrate the theories and provide a deeper understanding of the factors that contributed to adoption and continuance of online grocery shopping.

## Supporting information

**S1 File. Appendix.**
(DOCX)

**S2 File. Data.**
(XLSX)

## Author Contributions

**Conceptualization:** Viktoriya Galushko.

**Data curation:** Viktoriya Galushko.

**Formal analysis:** Viktoriya Galushko, Alla Riabchyk.

**Funding acquisition:** Viktoriya Galushko.

**Methodology:** Viktoriya Galushko, Alla Riabchyk.

**Writing – original draft:** Viktoriya Galushko.

**Writing – review & editing:** Viktoriya Galushko, Alla Riabchyk.

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
