## [Decision Letter · Decision Letter 0]

29 May 2023

PONE-D-23-11349The demand for online grocery shopping: COVID-induced changes in grocery shopping behaviour of Canadian consumersPLOS ONE

Dear Dr. Galushko,

Thank you for submitting your manuscript to PLOS ONE. After careful consideration, we feel that it has merit but does not fully meet PLOS ONE’s publication criteria as it currently stands. Therefore, we invite you to submit a revised version of the manuscript that addresses the points raised during the review process.

We look forward to receiving your revised manuscript.

Kind regards,

Vincenzo Basile, PhD

Academic Editor

PLOS ONE

Journal Requirements:

"This research was supported by the Deans Research Award, provided by the Faculty of Arts at the University of Regina to Viktoriya Galushko"

4. Thank you for stating the following in your Competing Interests section: "No authors have competing interests"

6. We note that you have referenced (ie. Davis, F. D. [1986]) which has currently not yet been accepted for publication. Please remove this from your References and amend this to state in the body of your manuscript: (ie Davis, F. D. [Unpublished]”) as detailed online in our guide for authors

**Additional Editor Comments:**

Please provide a final paper with all revisions made and I recommend an additional check on plagiarism and/or compliance with the Journal's guidelines.

Reviewers' comments:

Reviewer's Responses to Questions

**Comments to the Author**

1. Is the manuscript technically sound, and do the data support the conclusions?

Reviewer #1: No

Reviewer #2: Yes

2. Has the statistical analysis been performed appropriately and rigorously? 

Reviewer #1: No

Reviewer #2: Yes

3. Have the authors made all data underlying the findings in their manuscript fully available?

Reviewer #1: Yes

Reviewer #2: Yes

4. Is the manuscript presented in an intelligible fashion and written in standard English?

Reviewer #1: Yes

Reviewer #2: Yes

5. Review Comments to the Author

Reviewer #1: The paper aims to examine changes in online grocery shopping for Canadian consumers.

My major concern is on that the research results have very limited contribution to the theory and practice in the considered field. In its current format, the manuscript includes a limited set of references and simple analyses to address a complex topic. I believe that the manuscript would highly benefit from a reconfiguration of the data analyses frameworks (for more complex techniques, such as SEM, ANN…) to add value to this topic.

I advise improving on these aspects:

1. From the start, the manuscript’s title “The demand for online grocery shopping: COVID-induced changes in grocery shopping behaviour of Canadian consumers”, the idea of ‘changes’ it can induce the idea of longitudinal research with regard to grocery shopping before and after the pandemic.

Introduction

2. To start with, the abstract could present the original value of the paper.

3. From my perspective, the background of the manuscript is not portrayed in a comprehensive manner as it includes older statistics.

4. The academic background of the paper is not prominent in this section. The manuscript should focus on providing additional current, relevant, and academic sources.

5. Most importantly, the Introduction of the paper should also clearly address the aims, topicality, original value, and the research gaps addressed by this research.

6. The paragraph starting with: “Stemming from the results of a survey of more than 600 Canadian consumers” does not belong in the Introduction.

Background literature and theoretical framework

7. Overall, the literature review does not reflect thoroughness. From the beginning of this section, the manuscript mentions a classification of risks, but the idea is not followed through. Then, the section titled ‘Purchases during pandemics and in times of crisis’ is very briefly presented without enough substance to add a particular contribution. At the end of this section the manuscript provides statistics from Taiwan and UK, without an apparent link to the study context. Not enough substance and attention to details is provided.

8. Moreover, the following section “Consumer adoption of online grocery shopping: brief literature review” has the same flaws in presenting minimal details regarding “theory of reasoned action and the theory of planned behaviour [21], the technology acceptance model [22, 23], the theory of adoption of innovations [24, 25], and the perceived risk theory [26].”. These chosen theories seem randomly selected because there are multiple others that apply in the proposed research setting (e.g., UTAUT1/2)

9. In ‘Theoretical framework’ the manuscript discusses various ideas without providing actual theoretical perspectives or using any references. Again, in this section, the manuscript has phrasing consistent with longitudinal studies (eg. Paragraph starting with “Prior (pre-COVID) behaviour such as the frequency of in-store visits and general attitude”), whereas this manuscript’s simple analysis does not portray research design.

10. The manuscript mentions “The existing literature has shown that gender and age are two important descriptors of psychological pre-disposition to shop online.” However, no references are provided for this assessments. This is just one example of referencing previous research without providing relevant sources.

11. Based on ‘Figure 1. Theoretical model of the COVID-induced demand for online grocery purchases’, the manuscript seems to portray a SEM-based empirical research, whereas the actual analysis is based on much too simple types of data analyses. Thus, figure 1 is misleading in terms of providing a theoretical model which is not essentially applied or verified.

12. A major flaw of this section is related to the portrayal of the theoretical framework which does not emphasize cohesiveness, valuable, and comprehensive insights. In fact, the manuscript does not address a clear set of hypotheses to be examined and verified in a comprehensive manner. Additionally, the paper offers a very limited set of only 40 references and there are many more studies on this topic.

Research Methodology

13. Based on the available details in this section, I would assume that the scale items were not extracted from existing literature and appear to be newly developed. Thus, this raises the question of validity of the research instrument.

Results

14. Considering very basic data analysis techniques, highlighting only descriptive statistics and probit, the research results have very limited contribution to theory and practice. My major concern is related to the relevancy and topicality of this study in enhancing understanding consumers’ changing behaviors.

15. With regard to the application of probit, the manuscript fails to address any normality tests. Thus, the statistical analysis has not been performed appropriately and rigorously.

16. The manuscript mentions: “The results support our hypothesis that pre-COVID shopping behaviour played an important role in explaining why some consumers switched to online and some didn’t.” No formally phrased hypotheses were provided based on theoretical background and again “pre-COVID” implies examining changes in a longitudinal framework.

Conclusions and policy implications

17. The paper fails to address the original and relevancy of the paper. This is a major concern that I have highlighted based on my previous assessments.

18. For academic transparency, the final section needs to be restructured in its entirety to reflect the paper’s value, marketing implications for practical strategies, study limitations, and other possible ways to expand this study.

19. On a different note, there are certain instances in which the manuscript reflect informal phrasing.

I hope these recommendations have some value to the Author/s and their paper!

Reviewer #2: I have reviewed the article “A Timely Analysis of COVID-Induced Changes in Canadian Consumers' Grocery Shopping Behavior” PONE-D-23-11349

Review comments:

Introduction:

The authors present a comprehensive analysis of the impact of COVID-19 on the grocery shopping behavior of Canadian consumers, with a focus on the growing demand for online grocery shopping. The introduction should sets a strong foundation by clearly outlining the purpose and scope of the study.

Introduction.

Problem statement/ motivation is either missed or not efficiently communicated. It is suggested to

simplify the introduction in a way that the reader can understand the following Background of the study

Problem statement/motivation,Objectives of the study, Contribution of the study and Structure of the manuscript.

Methodology:

The study employs a rigorous methodology, combining quantitative data analysis and qualitative insights through surveys and interviews. The sample size and demographics are well-represented, ensuring the findings are generalizable to the Canadian population. The detailed explanation of the research design enhances the credibility of the study.

Research Methodology

Data

The primary data for this study was gathered from Indian customers. The demographic profile should be more specific. The inferences of the study may not adequately represent the population If it is from the entire population of the country. The study uses s 651 usable survey response. Population, sample and sampling need to be more elaborative and inclusive.

Results and Analysis:

The results presented in this study offer valuable insights into the changing grocery shopping behavior of Canadian consumers during the COVID-19 pandemic. The analysis is thorough and effectively highlights the key trends and patterns observed. The inclusion of statistical measures and visual representations aids in the interpretation of the data.

Online Grocery Shopping:

The authors shed light on the significant surge in demand for online grocery shopping among Canadian consumers due to the pandemic. They provide compelling reasons for this shift, including health concerns and convenience. The discussion is well-supported with relevant literature and industry examples, strengthening the overall argument.

Discussion

Currently the discussion is all about the relationship stated in the theoretical model which is already available in previous literature. However, the discussion in the context of grocery products is missing which is the core of the study (originality). Hence, it is suggested to enrich the discussion by accommodating more discussion that is online with the grocery products

Implications and Recommendations:

The study offers practical implications for both grocery retailers and policymakers. The authors emphasize the need for retailers to adapt to the increased demand for online shopping by improving their digital infrastructure and delivery services. The recommendations are actionable and aligned with the findings, making them valuable for stakeholders in the grocery industry.

.

Managerial implications

Please consider the same suggestion provided for the discussion and findings for the managerial

implications section as well.

6. Quality of Communication: Does the paper clearly express its case, measured against the technical language of the field and the expected knowledge of the journal readership? Has attention been paid to the clarity of expression and readability, such as sentence structure, jargon use, acronyms, etc. I suggest the author(s) to have English editing on the paper thoroughly, and I consider this as an essential factor. Proofreading is essential for this paper.

Limitations and Future Research:

While the study effectively captures the changes in grocery shopping behavior, it acknowledges certain limitations. The authors mention the potential for response bias and the inability to capture the long-term impact of the pandemic. They provide suggestions for future research, such as exploring the impact on different demographic groups and analyzing post-pandemic behavior.

Conclusion:

The conclusion effectively summarizes the main findings and reiterates the significance of the study. The authors conclude with a strong call to action, emphasizing the importance of adapting to the changing landscape of grocery shopping to meet the evolving needs of consumers.

Overall Assessment:

"The demand for online grocery shopping: COVID-induced changes in grocery shopping behavior of Canadian consumers" is an informative and timely study that contributes to the growing body of research on the impact of the pandemic on consumer behavior. The methodology is robust, the analysis is comprehensive, and the implications are practical. The paper is well-written and organized, making it accessible to both academic and industry audiences. I highly recommend this study for anyone interested in understanding the changing dynamics of grocery shopping in the Canadian market.

6. PLOS authors have the option to publish the peer review history of their article (what does this mean?). If published, this will include your full peer review and any attached files.

Reviewer #1: No

Reviewer #2: **Yes: **Sufyan Habib

---

## [Author Response · Author response to Decision Letter 0]

6 Aug 2023

Please, see the attached word file as it provides a better view of the responses. 

1. Comment 1: major concern. 

Response: We have considerably expanded our knowledge of the existing literature and the set of references. We have added references to more than 60 articles [in addition to the references we had in the original submission], including references of the studies discussing the key theoretical models of consumer behaviour as well as more recent studies exploring the impact of the COVID-19 pandemic on consumer buying behaviour. As a result, we believe that the revised manuscript now has a much stronger review of the background literature and a much stronger theoretical base for the empirical analysis. While we do agree that the paper would highly benefit from a reconfiguration of the data analysis framework, more specifically, addressing the topic via SEM, unfortunately, the existing dataset (the existing collected survey data) is missing some key questions/respondents’ responses that would be required for validation and measurement of some of the key constructs. This weakness of the existing dataset makes application of SEM technique impossible. Given the existing data we can only use one equation econometrics (Logit) estimation technique. We did add a mention of this weakness in our conclusions and one of the directions for our future research can be an extension of this study that would involve a new survey design and new data collection to ensure that we can explore and validate via SEM the multiple models of consumer behaviour that have been referenced in this work to set up the foundation for the Logit regression model. 

2. Comment 2: the idea of “changes” and how this induces the idea of longitudinal research.

Response: While the survey was administered at one point in time, in January 2022, meaning that we did not collect longitudinal data, the survey was administered almost 2 years after the onset of the pandemic. At the time when the survey was being administered, COVID was still a big issue (Canada was starting a new Omicron wave with the confirmed case numbers increasing rapidly); some of the government imposed restrictions (e.g. social distancing, mandatory masking, etc.) and policies were still in place. The survey questions explicitly asked participants to describe their grocery shopping behaviour prior to the pandemic, at the onset of the pandemic, and “currently” [i.e. January 2022]. Since the survey was not run too early into the pandemic (e.g. summer 2020), it is believed that in January 2022 consumers were fully aware of their “complete” adaptation to the pandemic and all the changes in buying behaviour the consumers had to make. So, although the data were not collected in a longitudinal manner, given the timing of the survey and the nature of the survey questions we were able to explore the COVID-induced “changes”. In the “Survey design” section we added a brief paragraph highlighting the fact that the questions were set up in such a way that consumers could report the changes in their grocery buying habits at different points in time. 

3. Comment 3: Introduction

Response: We have revised Introduction to clearly articulate the aim, the importance, and the research contribution of the study. While many research papers briefly mention the data and the findings in the introduction, we agreed with the first reviewer’s suggestion and removed the paragraph that briefly mentioned the survey data and the findings. Instead, we focused on highlighting the contribution of the paper to the existing research. 

4. Comment 4: Background literature and theoretical framework

Response: While in the original submission we tried to keep everything concise, having seen the first reviewer’s comment we were in full agreement that in many instances we presented the information in a very brief way. We have now created two separate sections. Section 2 presents Background literature, where we provide a brief review of the literature on consumer behaviour during pandemics in general and during the COVID pandemic in particular. This section now includes many recent studies on adaptations in consumer behaviour during COVID and provides a smooth link to a switch to online grocery shopping – the research interest of the paper – as one aspect of these adaptations. 

Section 3 uses the existing literature to build a theoretical model that focuses on consumer adoption of online grocery shopping; in this section we now clearly articulate the hypotheses (propositions). We have tried to provide theoretical perspectives and references to any claims (ideas) that are discussed. 

Note that we scraped the previous “Background literature and theoretical framework” and have completely re-wrote this section that is now Section 2 and Section 3 of the revised manuscript. We do agree that this section was previously under-referenced and the results of more than 60 additional studies have been added to support the discussion in Section 2 and 3. 

5. Comment 5: Research methodology.

Responses: Most of the questions in our survey questionnaire intended to elicit the changes that consumers had to make in response to the restrictions posed by the pandemic and to see which of these changes will have a long-lasting life beyond COVID. Since we had a slightly different goal in mind, we did not set up the questionnaire to include the scale items for multiple constructs to conduct a SEM analysis using the data. This is the weakness of this dataset and the reason why SEM technique can’t be applied to these data as mentioned above. 

6. Comment 6: Results

Response: Since most of the variables in our dataset are either dichotomous or ordered categorical, we agree with the first reviewer that the use of Probit cannot be justified as the residuals are likely not to be normally distributed. We have re-estimated the regression model using Logit, which is not built upon the assumption of normal distribution of the error term. The estimated (updated, Logit based) coefficients are similar (in terms of direction of the impact and its significance) to what we had before using Probit. We made the changes to our discussion of the results to reflect the new estimates and the significance of the impacts. We also expanded our discussion on certain aspects of adoption/expansion of online grocery shopping.

We hope that now that we have completely re-written the theoretical model section and explicitly stated the propositions (hypotheses), the reviewer will find that the discussion in the results section aligns much better with the theoretical model. 

7. Comment 7: Conclusions and policy implications

Response: We have made the appropriate adjustments to the concluding section of the paper to address the reviewer’s comments. 

8. Comment 8: Suggestion to have English editing on the paper

Response: The paper has been professionally edited (proofread by a professional editor specializing in Economics) before resubmission to ensure proper use of language throughout the paper.

---

## [Decision Letter · Decision Letter 1]

31 Aug 2023

PONE-D-23-11349R1The demand for online grocery shopping: COVID-induced changes in grocery shopping behaviour of Canadian consumersPLOS ONE

Dear Dr. Viktoriya Galushko,

Thank you for submitting your manuscript to PLOS ONE. After careful consideration, we feel that it has merit but does not fully meet PLOS ONE’s publication criteria as it currently stands. Therefore, we invite you to submit a revised version of the manuscript that addresses the points raised during the review process.

*Please provide a final paper with all revisions made and I recommend an additional check on plagiarism and/or compliance with the Journal's guidelines.*

We look forward to receiving your revised manuscript.

Kind regards,

Vincenzo Basile, PhD

Academic Editor

PLOS ONE

Reviewers' comments:

Reviewer's Responses to Questions

**Comments to the Author**

1. If the authors have adequately addressed your comments raised in a previous round of review and you feel that this manuscript is now acceptable for publication, you may indicate that here to bypass the “Comments to the Author” section, enter your conflict of interest statement in the “Confidential to Editor” section, and submit your "Accept" recommendation.

Reviewer #1: (No Response)

Reviewer #3: (No Response)

2. Is the manuscript technically sound, and do the data support the conclusions?

Reviewer #1: No

Reviewer #3: Yes

3. Has the statistical analysis been performed appropriately and rigorously? 

Reviewer #1: No

Reviewer #3: Yes

4. Have the authors made all data underlying the findings in their manuscript fully available?

Reviewer #1: (No Response)

Reviewer #3: No

5. Is the manuscript presented in an intelligible fashion and written in standard English?

Reviewer #1: Yes

Reviewer #3: Yes

6. Review Comments to the Author

Reviewer #1: The manuscript presents many new additions and in certain areas, improvements. However, there are many more aspects that are unclear. Moreover, the manuscript still lacks cohesiveness in ideas’ presentation. There seems to be many objectives and aims associated with the paper, however these aims are not portrayed and described in a consistent and cohesive manner.

Introduction and Literature Review

1. The research gaps and clear aims of the manuscript are still not portrayed. As the manuscript showcases multiple aims, these aims should be explained and supported throughout the text of the manuscript.

2. The paper lacks a comprehensive engagement with the existing literature on online grocery shopping behavior, particularly in the Canadian context. The new version of the manuscript highlights many more additional studies, however not all of them are addressing the literature framework in a relevant manner. A more robust theoretical framework is needed to underpin the research.

3. Integrating relevant theories from consumer behavior and e-commerce would enhance the paper's theoretical foundation. As previously mentioned, the manuscript lacks mentions of relevant papers to support theoretical arguments: ““Self-efficacy, in turn, will influence consumer’s perceived ease of use – a central concept in TAM. In the context of online grocery shopping, we postulate that self-efficacy is affected by technological abilities and opportunities to engage in online shopping and grocery shopping habits prior to the pandemic.”

4. Certain newly established proposals seem redundant (e.g., 2.c.) and others lack substantial theoretical basis for their development. In certain cases (e.g. propositions 7-10) seem to be combined without sufficient theoretical anchoring. This leads to lack of cohesiveness and substance. Also, in certain cases the phrasing leads the reader to anticipate SEM.

Methodology

5. Provide a comprehensive breakdown of the logit model's application, detailing variable selection, data collection methods, and model validation procedures. This clarity will enhance the paper's methodological rigor.

6. For the methodology, the paper raises major questions and concerns. The scale items are still not provided in this new version of the manuscript. However, at least a part of the scale items are presented in the Supplementary Material – Appendix. Nonetheless, this raises additional concerns about the included questions because we cannot understand the premises of the selection questions. Please offer additional details in section 4.3. Survey questions

7. The manuscript does not offer a profile of respondents. This is important because of the developments of the Models, particularly for the ‘Children_ages_6-11’ and ‘Children_under 6’ elements included in the models. Did all respondents have children? Please provide a comprehensive table reflecting the profile of the respondents in section 4.2.

8. Please address the chosen methods and offer examples of other studies that have included the same data analysis strategy. Thus, the rationale behind selecting these methods should be clearly stated. Why were these methods deemed appropriate for investigating COVID-induced changes in online grocery shopping behavior? This justification would strengthen the research's methodological foundation.

Results

9. Particularly with regard to section 5.1. and 5.2., it is difficult to link them to research clear objectives. The manuscript should present the results more comprehensively, offering context and meaningful insights, and link them explicitly to the research objectives.

10. What is the Number of valid observations for each model? These details should be presented in the tables.

11. The results highlighting ‘personality type’ are not comprehensively explained.

12. The manuscript does not offer insights into the assessment of the goodness-of-fit of the models.

13. The statements related to education and TAM mentioned across the manuscript, need additional contextualization and academic support. Such an example is: “Consistent with the TAM and TCT that postulate that more educated individuals are more likely to adopt new technologies”

14. While the addition of the new analysis presented in 5.3.2 (regarding the perceived advantages and disadvantages) is appreciated, this analysis lacks coherence with theoretical background and clear interpretations.

15. The interpretation of results is limited, with minimal discussion. The research results have very limited contribution and perceived significance to the theory and practice in the considered field. It is essential to focus on providing more comprehensive insights into the descriptive statistics and the model's findings, explaining their relevance to the research objectives.

Best of luck with the manuscript!

Reviewer #3: This is a manuscript on online grocery shopping behavior of Canadian consumers. The topic is timely and relevant, and the paper has merit. This is a revised version of the manuscript, and I was not one of the original reviewers. I read the response letter to the original reviewers. In my opinion, the authors have addressed the vast majority of the reviewers’ queries. From my own reading of the manuscript, I also feel it meets the required standards to warrant publication in the PLOS ONE journal.

I recommend implementing some additional (but very minor) changes, particularly by citing other recent work on the topic and discussing these new papers’ findings in the literature review section. I suggest citing at least two additional papers, but more could be added. Both papers are particularly relevant in this context and address another important market (Germany):

Brüggemann, P., & Olbrich, R. (2022). The impact of COVID-19 pandemic restrictions on offline and online grocery shopping: New normal or old habits?. Electronic Commerce Research (advance online publication). https://doi.org/10.1007/s10660-022-09658-1

Gruntkowski, L., & Martinez, L.F. (2022). Online grocery shopping in Germany: Assessing the impact of COVID-19. Journal of Theoretical and Applied Electronic Commerce Research, 17(3), 984-1002.

https://doi.org/10.3390/jtaer17030050

I wish the author(s) the best of luck with developing this line of research.

7. PLOS authors have the option to publish the peer review history of their article (what does this mean?). If published, this will include your full peer review and any attached files.

Reviewer #1: No

Reviewer #3: No

---

## [Author Response · Author response to Decision Letter 1]

8 Nov 2023

First, we would like to thank the reviewers for their insightful comments as we believe that addressing these comments has significantly improved the quality of the paper. Here, we briefly outline the revisions made to address the reviewers’ concerns/suggestions; the full extent of the revisions is obvious from the resubmitted manuscript that highlights the changes made via the track changes word feature. 

1. We have substantially expanded the Results section to provide a better link between the section developing the theoretical model and the sections explaining the results. We have improved the statistical analysis of the data as well as have provided information on a profile of respondents, by key consumer characteristics. 

2. We added the suggested references to better link the findings from this Canadian study to the findings from some other recent studies. Unfortunately, the literature on online grocery shopping behavior in the Canadian context is very scarce so we could not expand discussion in this front. 

3. We’ve also added references to some studies that used a similar approach, more specifically, logistic regression analysis, to analyze consumer adoption of e-commerce. 

Summarizing, we would like to thank the reviewers for their time providing such detailed and valuable comments. We believe that, in the current revised version of the manuscript, we have addressed the reviewers’ comments fully and hope that the reviewers will find the revised manuscript to be of a better quality, worth of publication in such a high impact journal as PLOS ONE (Economics).

---

## [Editor Report · Decision Letter 2]

13 Nov 2023

PONE-D-23-11349R2The demand for online grocery shopping: COVID-induced changes in grocery shopping behaviour of Canadian consumersPLOS ONE

Dear Dr. Viktoriya Galushko,

Thank you for submitting your manuscript to PLOS ONE. After careful consideration, we feel that it has merit but does not fully meet PLOS ONE’s publication criteria as it currently stands. Therefore, we invite you to submit a revised version of the manuscript that addresses the points raised during the review process.

*Please provide a final paper with all revisions made and I recommend an additional check on plagiarism and/or compliance with the Journal's guidelines.*

We look forward to receiving your revised manuscript.

Kind regards,

Vincenzo Basile, PhD

Academic Editor

PLOS ONE

Journal Requirements:

*We will check with the reviewers that the comments meet the quality standards for publication on PONE.:*

---

## [Author Response · Author response to Decision Letter 2]

22 Nov 2023

Dear Editors,

First we would like to thank you for providing comments on our resubmission in such a timely manner and for giving us an opportunity to resubmit our manuscript after addressing your minor comments. 

As per the journal requirements, we have carefully reviewed the reference list to ensure that it is complete and correct. In addition to correcting typos and completing the references with issue numbers where necessary, we have made the following changes to the reference list:

1. We moved references #113 and #114 to #59 & #60 as we felt that it was appropriate to mention these research works earlier in our manuscript (please see some edits on page 11 of the revised manuscript to incorporate these references). 

2. We replaced reference in #72 to Hochbaum et al. (1952), which is not an academic journal publication, with an academic journal publication. 

3. At the time of our 2nd revision, Reference #17 was published online but wasn’t published in a journal issue yet; the issue (issue No. 11) was just published in November 2023 and we updated the reference with the volume and issue. 

4. We have also added DOIs for the articles published in academic journals. 

We checked the correctness of the citations in the text as well as reviewed the reference list to ensure that the reference style is consistent with the PlosOne requirements for all works cited. We have also checked all the website links in the reference list to ensure that all of the links are still active. 

We’ve also made some minor edits in the text; more specifically, we have proof-read the whole manuscript again to eliminate any typos that were still present. 

Again, thank you very much for giving us an opportunity to resubmit our manuscript. 

Sincerely,

Viktoriya Galushko

---

## [Editor Report · Decision Letter 3]

24 Nov 2023

The demand for online grocery shopping: COVID-induced changes in grocery shopping behaviour of Canadian consumers

PONE-D-23-11349R3

Dear Dr. Viktoriya Galushko,

We’re pleased to inform you that your manuscript has been judged scientifically suitable for publication and will be formally accepted for publication once it meets all outstanding technical requirements.

Kind regards,

Vincenzo Basile, PhD

Academic Editor

PLOS ONE

*Please provide a final paper with all revisions made and I recommend an additional check on plagiarism and/or compliance with the Journal's guidelines.*

---

## [Editor Report · Acceptance letter]

31 Jan 2024

PONE-D-23-11349R3 

PLOS ONE

Dear Dr. Galushko, 

I'm pleased to inform you that your manuscript has been deemed suitable for publication in PLOS ONE. Congratulations! Your manuscript is now being handed over to our production team.

Kind regards, 

on behalf of

Dr. Vincenzo Basile 

Academic Editor

PLOS ONE